# The balance between B55α and Greatwall expression levels predicts sensitivity to Greatwall inhibition in cancer cells

The Greatwall kinase inhibits PP2A-B55 phosphatase activity during mitosis to stabilise critical Cdk1-driven mitotic phosphorylation. Although Greatwall represents a potential oncogene and prospective therapeutic target, our understanding of the cellular and molecular consequences of chemical Greatwall inactivation remains limited. To address this, we introduce C-604, a highly selective Greatwall inhibitor, and characterise both immediate and long-term cellular responses to the chemical attenuation of Greatwall activity. We demonstrate that Greatwall inhibition causes systemic destabilisation of the mitotic phosphoproteome, premature mitotic exit and pleiotropic cellular pathologies. Importantly, we show that the cellular and molecular abnormalities associated with reduced Greatwall activity are specifically dependent on the B55α isoform, rather than other B55 variants, underscoring PP2A-B55α phosphatases as key mediators of the cytotoxic effects of Greatwall-targeting agents in human cells. Additionally, we establish that sensitivity to Greatwall inhibition varies in different cell line models and that dependency on Greatwall activity reflects the balance between Greatwall and B55α expression levels. Our findings highlight Greatwall dependency as a cell-specific vulnerability and propose the B55α-to-Greatwall expression ratio as a predictive biomarker of cellular responses to Greatwall-targeted therapeutics.

Greatwall (GWL), also known as microtubule-associated serine/threonine-like (MASTL) kinase, promotes mitotic phosphorylation by suppressing PP2A-B55 activity[1,2]. Activated during the G2 phase by Cdk1, GWL phosphorylates two highly unstructured homologous proteins, ENSA and ARPP19[3,4]. Phosphorylated ENSA(S67) and ARPP19(S62) act as poor PP2A-B55 substrates and inhibit the Cdk1-counteracting PP2A-B55 complexes via an unfair competition mechanism[5]. The feedback loop involving Cdk1, GWL and PP2A-B55 integrates into the broader mitotic entry network and plays a crucial role in stabilising the phosphorylation of mitotic Cdk1 substrates[6]. This stabilisation contributes to the bistability and hysteresis of the G2/M transition and ensures unperturbed mitotic progression[7–9]. Accordingly, disruption of GWL function causes G2 arrest in *Xenopus laevis* egg extracts[10,11], prophase delays and chromosome condensation defects in developing *Drosophila Melanogaster* neuroblasts[12], and mitotic collapse followed by failed cytokinesis in mammalian cells[13–17]. In addition to its canonical role as the PP2A-B55 suppressor, GWL has been implicated in the modulation of metabolic homoeostasis via the mTOR and Akt pathways[18–22], regulation of DNA replication[23], DNA damage signalling[24] and promotion of Wnt/β-catenin signal transduction[24].

Beyond its fundamental cellular functions, GWL has emerged as a potential oncogene and therapeutic target[24–26]. Overexpression of GWL promotes tumorigenic characteristics, including chromosome instability, hyper-proliferation, loss of contact inhibition and partial epithelial-mesenchymal transition[21,22]. Accordingly, elevated GWL levels correlate with poor prognosis in breast, oral, gastric, colon, and head and neck cancers[27,28]. Depletion of GWL exhibits variable cytostatic effects in different breast cancer cell lines, indicating a

e-mail: r.zach@sussex.ac.uk; h.hochegger@sussex.ac.uk

therapeutic window for targeting cancer subtypes with increased GWL dependency[29]. Three independent groups have reported the development of small-molecule GWL inhibitors with variable inhibitory potential and antiproliferative effects in cancer cell line models[30–32] and mouse xenografts[33]. These reports, however, provide only minimal insight into the molecular and cellular mechanisms underlying the cytotoxicity of GWL inhibition, leaving several key questions unanswered. What are the molecular and cellular consequences of chemically attenuated GWL activity? Do cellular phenotypes associated with GWL inhibition differ in distinct cell types? Is the cytotoxicity of GWL inactivation purely driven by the deregulated PP2A-B55 phosphatases? And can cellular dependency on GWL activity be predicted?

In this study, we report the development of a highly selective GWL inhibitor, C-604, and present a comprehensive classification of cellular pathologies and phosphoproteomic changes associated with the chemical inactivation of GWL. We demonstrate that cytotoxic effects of GLW-targeting compounds originate from the untimely activation of PP2A-B55α phosphatases and consequential premature mitotic exit. Additionally, we characterise differential cellular dependencies on GWL activity and propose relative GWL and B55α expression levels as predictive biomarkers of sensitivity to GWL inhibition.

## Results

### Development of the GWL kinase inhibitor C-604
To discover small molecules interfering with GWL activity, we screened a kinase-focused subset of the AstraZeneca compound collection in the full-length hyperactive GWL (K72M) homogeneous time-resolved fluorescence (HTRF) assay. The kinase screening collection comprised around 40,000 samples of interest, augmented by exemplars of kinase hinge-binding motifs, literature-reported kinase actives and screening catalogue purchases (mean MW ≈ 360, mean clogP ≈ 2.8), aiming to achieve as wide a coverage of the kinome as possible. Initially, we selected potential active molecules with a z-score $\leq -5$ (equivalent to approximately 25% inhibition or better) and performed compound clustering, followed by near-neighbour testing and determination of relevant median inhibitory concentration ($IC_{50}$) values. Subsequent re-clustering based on GWL inhibitory potency, Cdk1 selectivity, broad kinome selectivity and additional known factors influencing drug developability, such as molecular weight, lipophilicity and the number of hydrogen donors and acceptors, generated six prioritised compound clusters (Supplementary Fig. 1a, b). Clusters 2, 4, and 6 exhibited poor selectivity towards Cdk1 and other kinases, disqualifying them from further drug development (Supplementary Fig. 1a, b). Cluster 1 ($n = 421$) exhibited a reasonable selectivity profile across a broad range of kinases; however, initial testing of 50 representative compounds indicated that eliminating the inhibitory activity against Cdk1 may prove challenging (Supplementary Fig. 1a, b). Despite the favourable kinase selectivity profile of cluster 5, we considered this series less attractive due to the limited number of compounds ($n = 5$) and suboptimal molecular properties for a hit, such as higher lipophilicity, lower binding efficiency (as indicated by ligand-lipophilic efficiency, LLE) and greater molecular weight (Supplementary Fig. 1a, b). Cluster 3 ($n = 20$) emerged as the most promising compound group, demonstrating an outstanding selectivity against Cdk1 and across a broader kinase panel (Supplementary Fig. 1a–c).

We selected compound **1** as the lead exemplar of cluster 3 and leveraged it as a foundation for subsequent drug development (Fig. 1a). Structure-activity relationship studies and pharmacokinetic profiling produced several derivative molecules, of which compound **21** provided the best combination of outstanding inhibitory potential, low clearance in mouse microsomes and acceptable solubility (Fig. 1a–c and Supplementary Tables 1–4). According to the HTRF-based $IC_{50}$ evaluation, both stereoisomers (S, R) of compound **21** exhibited equivalent inhibitory potential ($^{S/R}IC_{50} = 9.0 \pm 7.7$ nM; $^{S}IC_{50} = 10.9 \pm 2.0$ nM; $^{R}IC_{50} = 13.3 \pm 6.4$ nM), demonstrating that

compound **21** inhibited GWL in a non-stereospecific manner (Fig. 1c). This assessment was supported by structural modelling of the interaction between compound **21** and the active site of GWL (Fig. 1d). In further support of its GWL-targeting properties, compound **21** diminished the phosphorylation of the canonical GWL substrate ENSA(S67) in *Xenopus laevis* cytostatic factor (CSF) egg extract (Supplementary Fig. 2). Having satisfied the criteria of prospective GWL inhibitor, the racemic compound **21** was registered as UOS-00054604 and, for the remainder of the manuscript, is referred to as C-604.

### Compound C-604 interferes with GWL activity and inhibits proliferation in human cells
To determine whether C-604 inhibited GWL in human cells, we measured phosphorylation levels of canonical GWL substrates ENSA(S67) and ARPP19(S62) in prometaphase-arrested HCC1395 (breast cancer), U2OS (osteosarcoma), RPE-1 (non-transformed) and HeLa (cervical cancer) cells treated with C-604. We induced prometaphase arrest by treating the cells with S-trityl-L-cysteine (STLC) for 20 h and collected mitotic cells by mitotic shake-off (Fig. 1e, f). To minimise the impact of potential indirect effects, we analysed changes in ENSA(S67) and ARPP19(S62) phosphorylation levels 30 min after the administration of C-604 (Fig. 1f). In all four cell line models, exposure to increasing doses of C-604 resulted in gradual dephosphorylation of ENSA(S67) and ARPP19(S62) (Fig. 1g, h). Compound C-604 operated with a sub-micromolar median effective concentration ($EC_{50}$), implying outstanding potency (Fig. 1h). Importantly, estimated cellular $EC_{50}$ values ($^{HCC1395}EC_{50} = 0.28 \pm 0.21$ μM; $^{U2OS}EC_{50} = 0.43 \pm 0.15$ μM; $^{RPE-1}EC_{50} = 0.24 \pm 0.01$ μM; $^{HeLa}EC_{50} = 0.31 \pm 0.20$ μM) displayed only minor, statistically insignificant variations, indicating that C-604 acted consistently in different cellular models (Supplementary Fig. 3a).

A high dose of C-604 (2 μM) demonstrated strong cytostatic properties, effectively inhibiting colony formation in RPE-1 and HeLa cells exposed to the compound for 72 h (Fig. 1i, j). Interestingly, a lower concentration (0.5 μM) of C-604 partially inhibited proliferation in HeLa cells, but not in RPE-1 cells, suggesting a possibility of cell-specific sensitivities to C-604 treatment (Fig. 1i, j). A partial (c.a. 60%) knock-down of GWL by siRNAs significantly amplified the cytostatic potential of C-604 in both RPE-1 and HeLa cells, indicating a direct link between C-604-induced loss of viability and low GWL activity (Fig. 1i, j and Supplementary Fig. 3b, c). Notably, siRNA-mediated depletion of B55α, the dominant isoform of PP2A regulatory subunit B55, had the opposite effect and significantly rescued the proliferative capacity of C-604-treated RPE-1 and HeLa cells (Fig. 1i, j and Supplementary Fig. 3b–d). Considering that the untimely activation of mitotic PP2A-B55 phosphatases represents a known cytotoxic consequence of GWL inactivation[13,34], the observation that the antiproliferative activity of C-604 depended on B55α implied a selective targeting of the PP2A-B55/ENSA/GWL pathway. Our preliminary biological characterisation thus strongly suggested that C-604 inhibited GWL and displayed a strong PP2A-B55-dependent cytostatic potential in human cells.

### Greatwall inhibitor C-604 causes pleiotropic B55α-dependent cellular pathologies
Several independent studies have demonstrated that disruption of GWL activity by genetic approaches causes defective mitotic progression, with detrimental cellular consequences such as polyploidisation, cell cycle arrest, aberrant nuclear morphology and apoptosis[12–17]. To determine whether C-604 induced analogous physiological defects, we utilised high-throughput immunofluorescence microscopy combined with custom image analysis software to quantify fundamental characteristics of cell populations subjected to increasing doses of C-604. We employed a four-channel fluorescent staining procedure involving (1) immuno-labelling of α-tubulin, (2) immuno-labelling of the cell cycle inhibitor p21, (3) Click-iT-based EdU-labelling, and (4) DNA staining with Hoechst. To accurately segment

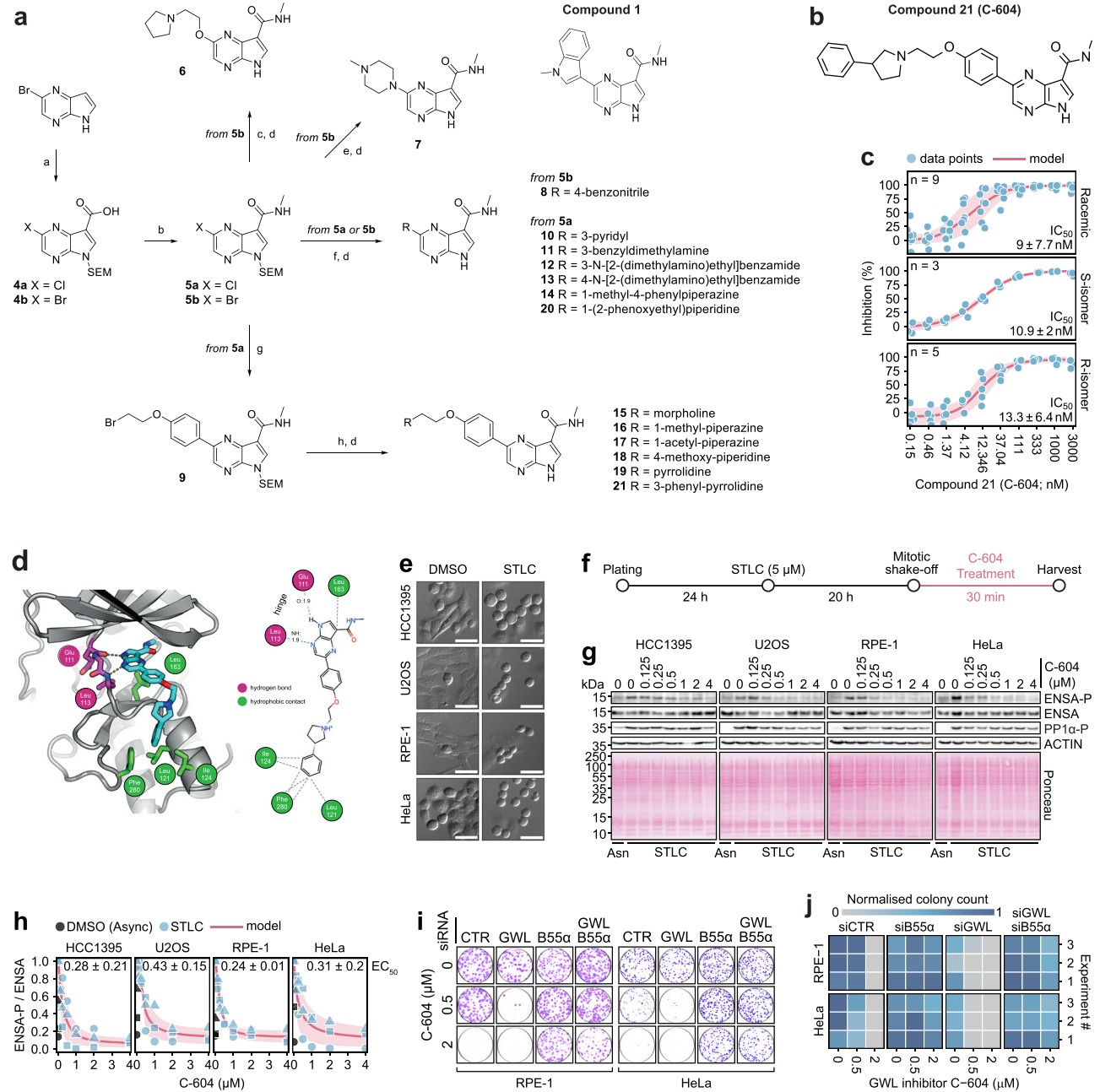

**a** (reaction scheme: Compound 1, Compound 21 (C-604), compounds 6, 7, 8, 10–21)

**b** Compound 21 (C-604)

**c** data points — model; Racemic n = 9, IC50 9 ± 7.7 nM; S-isomer n = 3, IC50 10.9 ± 2 nM; R-isomer n = 5, IC50 13.3 ± 6.4 nM; Compound 21 (C-604; nM)

**d** hydrogen bond; hydrophobic contact

**e** DMSO / STLC; HCC1395, U2OS, RPE-1, HeLa

**f** Plating — 24 h — STLC (5 µM) — 20 h — Mitotic shake-off — C-604 Treatment 30 min — Harvest

**g** HCC1395, U2OS, RPE-1, HeLa; ENSA-P, ENSA, PP1α-P, ACTIN, Ponceau

**h** DMSO (Async), STLC, model; EC50 values HCC1395 0.28 ± 0.21, U2OS 0.43 ± 0.15, RPE-1 0.24 ± 0.01, HeLa 0.31 ± 0.2

**i** siRNA: CTR, GWL, B55α, GWL B55α; C-604 (µM)

**j** Normalised colony count

---

cells and nuclei, we utilised the Cellpose segmentation algorithm[35] and custom cell-specific segmentation models (Fig. 2a). For each identified cell, we extracted key features, including cell area, nuclear count and intensities of DNA staining, EdU-labelling and p21 immuno-labelling (Fig. 2a). An assessment of cell areas and nuclear counts provided insight into general changes in cellular and nuclear morphology. Thresholding of integrated Hoechst intensities discriminated between distinct relative states of ploidy, including 2 N, 4 N and polyploid 8 N+ (Fig. 2b). Additionally, distributions of EdU and p21 intensities allowed the identification of EdU-positive (EdU+) S-phase cells and p21-positive (p21+) quiescent or senescent cells (Fig. 2b).

To capture potential cell-specific differences in responses to C-604, we analysed a diverse panel of cell lines representative of different solid tumour types, including HCC1395, U2OS, RPE-1, and HeLa, as well as two additional breast cancer cell lines, MDA-MB-231 (MM231) and BT-549 (Fig. 2c). Consistent with the reported outcomes of genetic GWL manipulations, chronic exposure to C-604 caused pleiotropic physiological abnormalities, including polyploidisation, upregulation

of p21, nuclear fragmentation and a significant increase in cell size (Fig. 2c, d and Supplementary Fig. 4). Notably, C-604-induced pathologies manifested differently in cancer-derived cell line models and non-transformed RPE-1 cells (Fig. 2c, d and Supplementary Fig. 4). Following a 72-hour exposure to C-604, cancer cells, including HCC1395, U2OS, MM231, HeLa and BT-549, retained replicative proficiency and accumulated large, often multinucleated, cells with 4 N or polyploid 8 N+ DNA content (Fig. 2c, d and Supplementary Fig. 4). Notably, the extent of nuclear and genomic deterioration varied, with HeLa and BT-549 cells showing the highest incidence of nuclear fragmentation and polyploidisation (Fig. 2c, d and Supplementary Fig. 4). C-604-treated cancer cells also exhibited varying levels of p21 expression; however, upregulation of this cell cycle inhibitor appeared to have a minimal impact on their replicative proficiency (Fig. 2c, d and Supplementary Fig. 4). C-604-treated non-transformed RPE-1 cells exhibited a robust p21 upregulation, the suppression of DNA replication and the accommodation of a senescence-like cellular state with 2 N or 4 N DNA content (Fig. 2c, d and Supplementary Fig. 4). Most C-

**Fig. 1 | Development and characterisation of the GWL kinase inhibitor C-604.** **a** Chemical structure of the lead exemplar compound 21 and the general synthetic route towards the preparation of pyrrolopyrazine GWL inhibitors. (a) for 4a, (i) NaH, SEM-Cl, DMF, 0 °C, 4 h; (ii) Arnold's reagent, CHCl₃, 60 °C, 20 h; (iii) NaClO₂, KH₂PO₄, NH₂SO₃H, 1,4-dioxane/water, 0 °C to rt, 2 h, 35% (three steps); for 4b, (i) hexamine, TFA, MW, 80 °C, 20 min; (ii) NaH, SEM-Cl, DMF, 0 °C to rt, 16 h; (iii) NaClO₂, KH₂PO₄, NH₂SO₃H, 1,4-dioxane/water, 0 °C to rt, 1.5 h, 14% (three steps); (b) 2 M methylamine in THF, T3P, pyridine, EtOAc, 60 °C, 2-4 h, 89-97%; (c) 2-(pyrro-lidin-1-yl)ethanol, CuI, L-proline, K₂CO₃, DMSO, 100 °C, 16 h, 48%; (d) TFA, DCM, rt, 16 h then ethylenediamine, DCM, rt, 3-6 h, 29-92%; (e) 1-methylpiperazine, CuI, L-proline, K₂CO₃, DMSO, 100 °C, 16 h, 77%; (f) boronate ester, Pd(dppf)Cl₂ · DCM, 1,4-dioxane/water, MW, 120 °C, 30 min, 43-81%; (g) boronate ester, Pd(dppf)Cl₂ · DCM, 1,4-dioxane/water, MW, 120 °C, 30 min, 56%; (h) amine, K₂CO₃, DMF, 95 °C, 1 h, 32-67%. **b** Chemical structure of the lead compound 21 (C-604). **c** Dose-dependent effect of compound 21 (C-604) on the in vitro GWL activity measured by an HTRF assay utilising purified full-length GWL (K72M) and ENSA. Per cent (%) inhibition was determined from fluorescence intensity values normalised to the signal measured in the control enzyme-free reaction. Red lines and shaded regions represent means and standard deviations of independently fitted four-parameter sigmoidal models. Dots represent individual measurements. Mean IC₅₀ values ± standard deviations are indicated. Numbers of independent experiments (n) are indicated. **d** Model engagement of compound 21 (C-604) with the GWL active site. Left, secondary structure molecular cartoon. Amino acid residues predicted to interact with

compound 21 are shown in stick representation, with carbon atoms coloured magenta or green if they make hydrogen bonds or hydrophobic contact with the compound, respectively. Right, 2D interaction map. Refer to the associated key for further details. **e** Representative differential interference contrast (DIC) images of DMSO-treated interphase cells and STLC-treated prometaphase-arrested cells. The scale bar represents 50 μm. **f** A schematic outline of the experimental procedure to establish cellular EC₅₀ values of the C-604 inhibitor. **g** Representative Western blots showing the effect of C-604 on ENSA(S67) and ARPP19(S62) phosphorylation. ENSA and ENSA-P refer to total and phosphorylated ENSA/ARPP19 levels, respectively. Phosphorylation of PP1 at T320 (PP1-P) represents a mitotic marker. The experiment was repeated n = 3 times. **h** Normalised ratios of ENSA-P and ENSA levels in prometaphase-arrested cells treated with C-604. Data were normalised to the maximum value in each given set. Dots, triangles and squares indicate the results of n = 3 independent experiments. Red lines and shaded regions represent means and standard deviations of n = 3 independently fitted four-parameter sigmoidal models. Mean C-604 EC₅₀ values ± standard deviations are shown for each tested cell line. **i** Colony formation capacity of RPE-1 and HeLa cells transfected with siRNAs targeting GWL or B55α. Cells were treated with C-604 for 72 h and subsequently grown in a drug-free medium for 7-10 days. Representative images from one of n = 3 experiments are shown. **j** Quantification of data presented in (e). In each experiment, cell-line-specific colony counts were normalised to the highest recorded value. The results of n = 3 independent experiments are shown as individual rows in the heatmap.

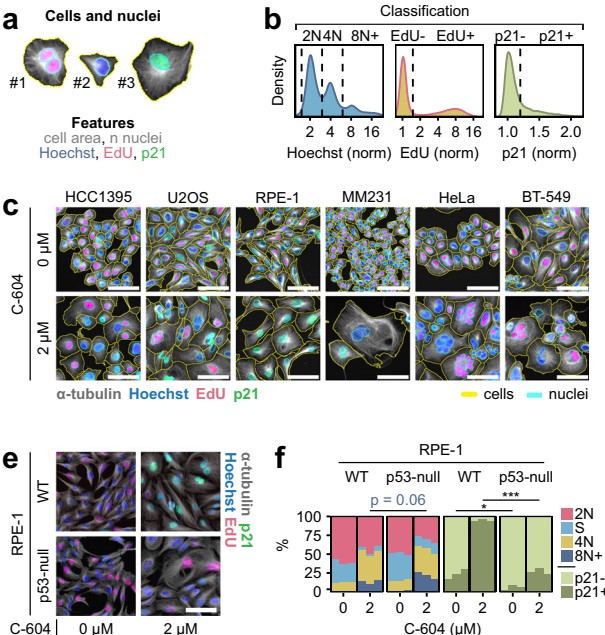

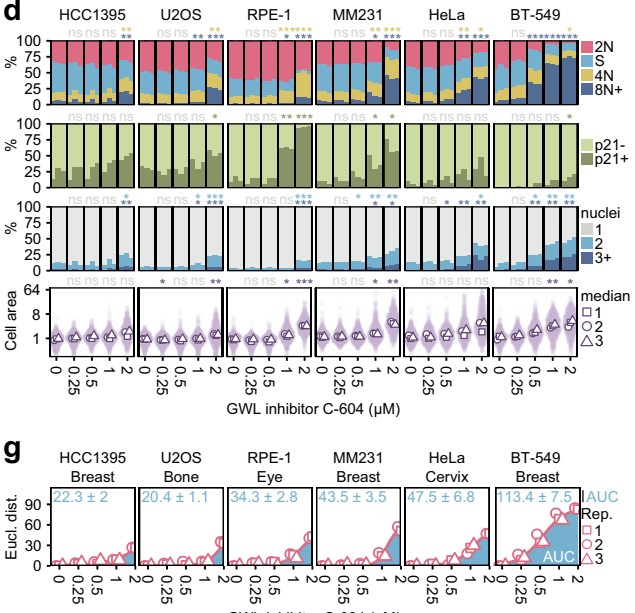

**Fig. 2 | Characterisation of long-term cellular consequences of GWL inhibition by C-604.** **a** Representation of the object and feature extraction utilising Cellpose segmentation[35]. **b** Demonstration of object classification based on distributions of Hoechst, EdU and p21 intensities. **c** Representative images of cells treated with C-604 for 72 h. Yellow and cyan outlines indicate segmented cells and nuclei, respectively. **d** Distributions of cell cycle groups, proportions of quiescent/senescent p21+ cells, frequency of polynucleated cells, and normalised cell areas in cell populations treated with C-604 for 72 h. The results of n = 3 independent experiments are shown for each condition as separate columns. The cell cycle state in the stacked bar plot and indication of statistical significance are colour-coded as indicated by the colour legend below the plots. Between 600 (Experiment 1, MM231, 2 μM C-604) and 43,495 (Experiment 3, HeLa, 0 μM C-604) cells were analysed. Statistical significance was determined using an unpaired two-tailed t-test from n = 3 independently determined proportions (%) or median values (cell area). In the cell cycle analysis, only differences in proportions of 4 N and 8 N+ cells were statistically tested. Parametric statistical testing was justified by the Shapiro-Wilk

test of normality (Supplementary Fig. 4c). Determined p-values were adjusted for multiple testing using the Benjamini-Hochberg procedure. The effect size analysis (Cohen's d) is presented in (Supplementary Fig. 4d). **e** Representative images of WT and p53-null RPE-1 cells treated with C-604. Images from one of n = 3 independent experiments are shown. **f** Distributions of cell cycle groups and proportions of p21+ cells in WT and p53-null RPE-1 cells treated with C-604. Between 511 (Experiment 3, RPE-1 p53-null, 2 μM C-604) and 7,754 (Experiment 2, RPE-1 WT, 0 μM C-604) cells were analysed. The results of n = 3 independent experiments are shown for each condition as separate columns. Statistical significance was determined from n = 3 independently determined proportions (%) using an unpaired two-tailed t-test. Parametric statistical testing was justified by the Shapiro-Wilk test of normality (p ≥ 0.15). **g** Euclidean distances between proportions of cell cycle groups in control and C-604-treated cell populations (d). Points represent the results of n = 3 independent experiments. Mean AUC values ± standard deviations are indicated. AUC – area under the curve, ns – not significant, * p < 0.05, ** p < 0.01, *** p < 0.001. The scale bars in (c, e) represent 50 μm.

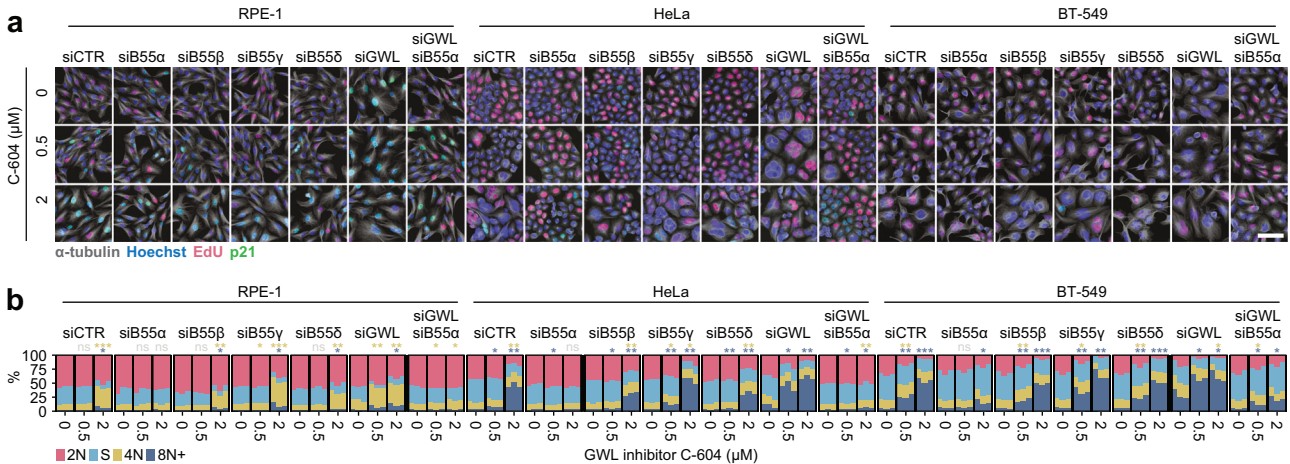

**Fig. 3 | The impact of siRNA-mediated depletion of GWL and B55α-δ on cellular responses to GWL inhibition by C-604. a** Representative images of RPE-1, HeLa and BT-549 cells treated with siRNAs targeting B55α, B55β, B55γ, B55δ or GWL exposed to C-604 for 48 h. Images from one of $n = 3$ experiments are shown. The scale bar represents 50 μm. **b** Distributions of cell cycle groups in control and C-604-treated populations depleted of B55α-δ or GWL. The results of $n = 3$ independent experiments are shown for each condition as separate columns. The cell cycle state in the stacked bar plot and indication of statistical significance are colour-coded as specified by the colour legend below the plots. Between 407 (Experiment 3, BT-549, siB55β, 2 μM C-604) and 32,941 (Experiment 3, HeLa, siB55β, 0 μM C-604) cells were analysed. Statistical significance of differences between proportions of 4 N and 8 N+ cells was determined by an unpaired two-tailed t-test. Parametric statistical testing was justified by the Shapiro-Wilk test of normality ($p \geq 0.46$). Determined p-values were adjusted for multiple testing using the Benjamini-Hochberg procedure. ns – not significant, * $p < 0.05$, ** $p < 0.01$, *** $p < 0.001$.

604-treated RPE-1 cells maintained an unperturbed nuclear architecture, with only a minor proportion displaying multinucleation (up to 16%) and polyploidisation (up to 11%) (Fig. 2c, d).

Considering that all tested cancer cell lines carried genetic perturbations negatively impacting p53 function (Supplementary Fig. 5a), we hypothesised that the fundamental differences in cellular responses to C-604 represented a direct consequence of defective p53 signalling. This premise was only partly supported by the experimental evidence, as C-604-treated RPE-1 cells lacking p53 (p53-null) showed significantly weaker induction of p21, did not fully inhibit DNA replication, but displayed only minor, statistically insignificant ($p = 0.06$), increase in polyploidisation frequency when compared with WT RPE-1 cells (Fig. 2e, f and Supplementary Fig. 5b, c). Other p53-independent pathways could thus be involved in shaping the physiological outcomes of chemical GWL inhibition.

Consistent with observed differences between cytostatic potential of C-604 in HeLa and RPE-1 cells (Fig. 1i, j), dose-dependent magnitudes of C-604-induced phenotypic alterations differed in distinct cell line models (Fig. 2d and Supplementary Fig. 4). To assess these differences quantitatively, we calculated Euclidean distances between proportions of different cell cycle groups (2 N, S, 4 N, 8 N + ) of control and C-604-treated cells (Fig. 2g). To approximate cell-specific C-604 sensitivities, we represented each Euclidean C-604 response profile by its area under the curve (AUC) (Fig. 2g). The calculated AUC values varied substantially, with more than a five-fold difference between the least responsive HCC1395 and U2OS cells and the most sensitive BT-549 cells (Fig. 2g). The observed variation in C-604 sensitivities could not be attributed to different histological origins of tested cell lines (Fig. 2g), foreshadowing the existence of other cellular mechanisms that influence the susceptibility to C-604-mediated GWL inhibition.

In agreement with the results of colony formation assays (Fig. 1i, j), reduction of GWL expression by GWL-specific siRNAs increased the general sensitivity to C-604 (Fig. 3 and Supplementary Fig. 6). In contrast, the siRNA-mediated depletion of B55α significantly suppressed all C-604-induced phenotypes across the experimental cell line panel (Fig. 3 and Supplementary Fig. 6). Notably, the siRNA-mediated targeting of other B55 isoforms B55β, B55γ, and B55δ had only a minimal impact on the phenotypic outcomes of the C-604 treatment,

highlighting a unique role of PP2A-B55α phosphatases in the disruption of cellular functions following GWL inhibition (Fig. 3 and Supplementary Fig. 6). In the case of B55β and B55δ, the absence of a rescue effect was unlikely a result of inefficient depletion, as siRNA transfections substantially reduced RNA expression of both isoforms (Supplementary Fig. 3d). The absence of a rescue effect in cells transfected with B55γ-specific siRNAs was consistent with previous reports indicating that B55γ expression is restricted to brain tissues[36]. Accordingly, B55γ RNA was undetectable even in cells transfected with control siRNAs (Supplementary Fig. 3d). These results strongly implied that C-604-induced cellular pathologies resulted from GWL inhibition and consequential de-repression of PP2A-B55α. In agreement with this interpretation, chemical inhibition of Cdk1, known to be counteracted by PP2A-B55 phosphatases, significantly increased the sensitivity to C-604 in all tested cell lines (Supplementary Fig. 7).

**Greatwall inhibitor C-604 acts on-target**

Although we demonstrated that C-604 induced PP2A-B55α-dependent cellular defects analogous to those associated with GWL depletion, supporting its on-target activity, we could not yet exclude the possibility of additional off-target effects. To identify potential C-604 off-targets, we utilised a commercially available off-target screening service (Thermo Fisher Scientific) to assess the impact of C-604 on the in vitro activity of 100 human kinases (Supplementary Fig. 8a). Out of 100 tested targets, C-604 (1 μM) inhibited GWL (inhibition 95%) and an additional 10 kinases (inhibition ≥ 80%). Of these 10 initial hits, C-604 exhibited strong inhibitory potential (in vitro $IC_{50} \leq 50$ nM) against NUAK1, MAP3K9, MELK, ULK2 and HIPK2 (Supplementary Fig. 8b, c). Besides these five kinases, we considered moderately impacted CHEK1 (inhibition 74%, in vitro $IC_{50} = 46.2$), the most prominent hit with a common essential function (Supplementary Fig. 8a–c).

To test whether the most prominent off-target suspects contributed to C-604-dependent cellular pathologies, we utilised immunofluorescence microscopy to analyse phenotypes of untreated and C-604-treated HeLa and RPE-1 cells transfected with siRNAs targeting GWL, NUAK1, MAP3K9, MELK, ULK2, HIPK2 and CHEK1 (Supplementary Fig. 8d-f). We reasoned that depletion of any genuine off-target would cause at least one of the three effects, including (1) full or partial

recapitulation of the C-604-induced phenotypes, (2) synergy with the low (0.5 μM) dose of C-604 and (3) no further amplification or alteration of the effects caused by the high dose (2 μM) of C-604. To evaluate these hypothetical scenarios, we analysed 500 randomly selected cells from each tested population, quantifying DNA staining intensity, EdU incorporation, p21 immunolabelling intensity, cell area, and nuclear count (Supplementary Fig. 8g). Then, we used the uniform manifold approximation and projection (UMAP) algorithm to project these five characteristics onto a two-dimensional space (Supplementary Fig. 8g, h). Finally, for each UMAP projection, we calculated the Wasserstein distance from the UMAP representation of cells transfected with the control siRNA (siCTR) and treated with a high dose of C-604 (2 μM) (Supplementary Fig. 8g-i). The low Wasserstein distance values indicated similarity and implied that the given siRNA treatment did not alter the character of phenotypes caused by C-604. In contrast, the high Wasserstein distance values suggested phenotypic differences consequential to the lack of detectable phenotype or distinct siRNA-specific effects. UMAP representations of cell populations depleted of NUAK1, MAP3K9, MELK, ULK2, HIPK2, and CHEK1 displayed increased Wasserstein distances in untreated populations as well as in cells exposed to low (0.5 μM) and high (2 μM) doses of C-604 (Supplementary Fig. 8h, i). These results indicated that the suppression of the suspected off-targets (1) did not replicate the effects of C-604, (2) did not synergise with the low dose (0.5 μM) of C-604, and (3) further modified phenotypes specific to the high dose (2 μM) of C-604. Notably, UMAP projections of HeLa and RPE-1 cells transfected with GWL-specific siRNAs showed the smallest Wasserstein distance values, both under normal conditions and in populations treated with low (0.5 μM) or high (2 μM) doses of C-604. (Supplementary Fig. 8h, i). Although we could not exclude the possibility of partial effects on other biological functions, these results strongly supported GWL as the primary target of C-604.

Overall, the initial characterisation of chronic physiological responses to C-604 treatment and the assessment of potential off-target effects supported the function of C-604 as a genuine GWL inhibitor and established PP2A-B55α phosphatase as the primary driver of cellular pathologies associated with chemical GWL inactivation. Notably, the presented results also raised important questions. First, how does C-604 compare with alternative GWL-targeting drugs? Second, what are the mechanistic origins of long-term C-604-induced pathologies? Third, what molecular processes are impacted by the chemical GWL inhibition? And fourth, what physiological features underlie the varying dependency on unperturbed GWL activity?

## Comparison of C-604 and the alternative GWL inhibitor MKI-2

To assess the efficacy of C-604 relative to other available GWL-targeting compounds, we compared the phenotypic effects induced by C-604 with those triggered by the alternative GWL inhibitor MKI-2, an improved derivative of its predecessor, MKI-1[31,32]. We focused our analysis on MKI-2, as it represented the most recently developed and independently characterised GWL inhibitor available at the time[31]. In contrast, other reported GWL inhibitors lacked sufficient biological characterisation[33].

We synthesised MKI-2 according to the published protocols[31] and determined its impact on ENSA(S67) and ARPP19(S62) phosphorylation levels in prometaphase-arrested HeLa and RPE-1 cells. Consistent with previous characterisation[31], 30 min exposure to MKI-2 resulted in a dose-dependent decrease in ENSA(S67) and ARPP19(S62) phosphorylation levels (EC$_{50}$ = 0.114 μM), indicating that MKI-2 interfered with GWL activity (Supplementary Fig. 9a–c). To evaluate the effects of MKI-2 on cellular physiology, we analysed phenotypes of MKI-2-treated HeLa and RPE-1 cells transfected with siRNAs targeting GWL or B55α (Supplementary Fig. 9d–g). MKI-2 induced evident cellular pathologies and suppressed proliferation in both analysed cell lines at a relatively low dose (0.031 μM) (Supplementary Fig. 9d, e). Notably,

the minimum cytostatic MKI-2 dose (0.031 μM) did not impact mitotic ENSA(S67) and ARPP19(S62) phosphorylation (Supplementary Fig. 9a–c), suggesting that the observed cytotoxicity may represent an off-target effect. In line with this assessment, 0.031 μM MKI-2 did not cause accumulation of 4 N or 8 N+ cells, a well-characterised consequence of low GWL activity[12–17]. (Supplementary Fig. 9f, g). Instead, the minimum cytostatic dose of MKI-2 arrested RPE-1 cells in a p21-positive 2 N state and induced only minor signs of polyploidisation in HeLa cells (Supplementary Fig. 9f, g). At this dose, MKI-2 caused accumulation of 4 N or 8 N+ cells only in cell populations transfected with GWL-specific siRNAs (Supplementary Fig. 9f). At higher concentrations (0.125–2 μM), MKI-2 induced a broad range of dose-dependent phenotypes, including S-phase inhibition and accumulation of 4 N or 8 N+ cellular states (Supplementary Fig. 9f, g). Importantly, siRNA-mediated depletion of B55α did not rescue MKI-2-induced pathologies across the entire range of tested concentrations (Supplementary Fig. 9d–g). The only effect of B55α-specific siRNAs was the attenuation of the elevated proportions of 4 N and 8 N+ cells induced by the combined action of GWL depletion and the minimal cytostatic concentration of MKI-2 (Supplementary Fig. 9f). Nonetheless, cells co-transfected with siRNAs targeting both B55α and GWL continued to exhibit pronounced proliferation defects and, in the case of RPE-1 cells, a marked upregulation of p21 (Supplementary Fig. 9d–g). Unlike the impacts of MKI-2, phenotypes associated with C-604 treatment, including inhibition of proliferation, accumulation of 4 N or 8 N+ cells and, in the case of RPE-1 cells, upregulation of p21, were entirely dependent on B55α (Supplementary Fig. 9d–g). These observations indicated that, unlike the effects of C-604, the primary cytotoxicity of MKI-2 did not represent a direct consequence of GWL inhibition. Consequently, we concluded that, although MKI-2 interfered with GWL activity, its limited specificity rendered it unsuitable for the reliable assessment of cellular GWL function. Since other available GWL inhibitors lacked comprehensive biological characterisation[33], C-604 emerged as the most thoroughly biologically validated GWL-targeting tool compound to date.

## Inhibition of GWL causes premature mitotic exit and cytokinetic failure

To determine the mechanistic origin(s) of long-term C-604-induced pathologies, we utilised live-cell imaging and analysed the immediate effect of C-604 in HCC1395, U2OS, MM231, RPE-1, HeLa and BT-549 cells expressing α-tubulin-GFP and H2B-mCherry. In all tested cell lines, administration of C-604 triggered premature mitotic exit, leading to mitotic slippage or defective cytokinesis (Fig. 4a). While mitotic slippage was relatively rare, cytokinetic failure represented the dominant outcome of the C-604 treatment (Fig. 4a). The inability to resolve newborn daughters typically caused an immediate or delayed collapse into a large polyploid body with an aberrant nuclear architecture (Fig. 4a). Alternatively, in RPE-1 cells, incomplete cytokinesis occasionally produced sister cells connected by an extended cytoplasmic bridge (Fig. 4a). Exposure to C-604 also extended mitotic timing; however, prolonged mitosis did not always precede mitotic catastrophe, suggesting that changes in mitotic duration represented an accompanying phenomenon rather than the cause of observed cell division defects (Fig. 4b). The occurrence of C-604-induced mitotic pathologies was dose-dependent and varied across distinct cell line models (Fig. 4c). Notably, varying frequencies of aberrant mitotic outcomes correlated with the established magnitudes of long-term C-604 effects, implying a direct link between the untimely mitotic exit and the broad spectrum of cellular defects associated with the long-term exposure to C-604 (Fig. 4d). Based on the visual inspection of generated movies, premature mitotic exit inhibited or considerably delayed subsequent mitotic commitments in HCC1395, U2OS, MM231, and RPE-1 cells (Fig. 4a). In contrast, HeLa and BT-549 cells readily attempted secondary, highly deleterious cell divisions characterised

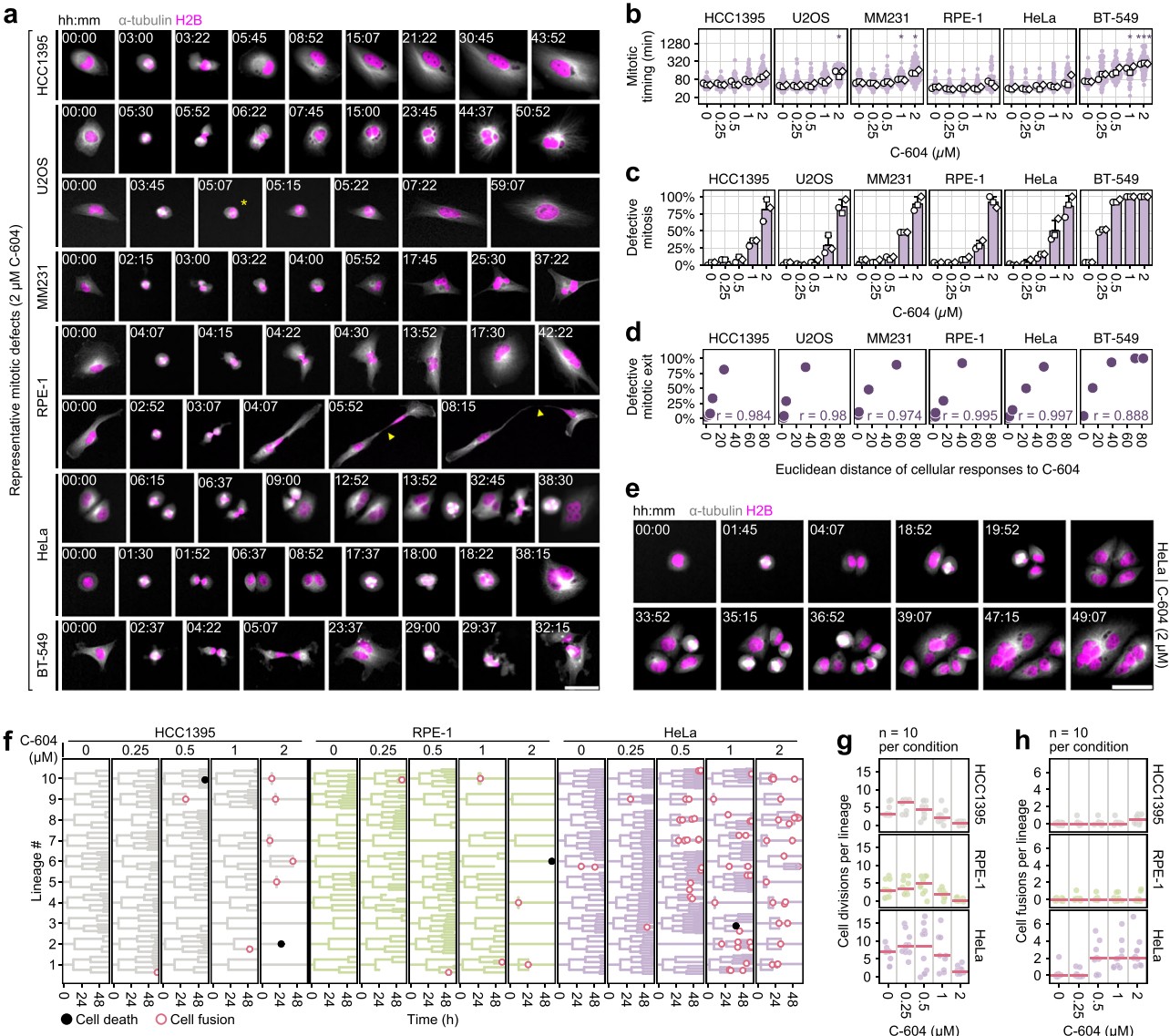

**Fig. 4 | Characterisation of mitotic consequences of GWL inhibition by C-604.**
**a** Representative time-lapse images of aberrant mitotic events and consequential cellular defects induced by C-604. Images from one of $n = 3$ experiments are shown. Arrowheads indicate a cytoplasmic bridge between newborn RPE-1 cells. The asterisk indicates mitotic slippage. **b** Impact of C-604 on mitotic timing determined as the distance between mitotic entry and the first signs of cytokinesis or mitotic slippage. Purple points represent pooled individual measurements. White points represent medians of $n = 3$ independent experiments (circle, square, rhombus). Statistical significance was determined from $n = 3$ medians, using an unpaired two-tailed t-test. Determined p-values were adjusted for multiple testing using the Benjamini-Hochberg procedure. * $p < 0.05$, ** $p < 0.01$, *** $p < 0.001$. **c** Frequencies of defective mitotic exit. White points represent the results of $n = 3$ independent

experiments. Bars and error bars represent means and standard deviations, respectively. (b, c) 25-50 mitotic events were analysed for each experiment, cell line and condition. **d** Correlation scatterplots of defective mitotic exit frequencies and magnitudes of cellular responses to C-604 treatment (Fig. 2g). Pearson correlation coefficients are indicated. **e** Representative image sequence demonstrating C-604-induced cellular fusions between cousin HeLa cells. An image sequence from one of three independent experiments is shown. **f** Cell lineages ($n = 10$) of HCC1395, RPE-1 and HeLa cells treated with increasing doses of C-604 for 60 h. Black and red circles represent cell death and fusion events, respectively. **g, h** Cell division (g) and fusion (h) events per lineage. Points represent individual lineages ($n = 10$ per condition). Horizontal red lines represent medians.

by the formation of multipolar mitotic spindles, extensive nuclear damage and further polyploidisation (Fig. 4a). Intriguingly, polyploidisation events in HeLa cells infrequently involved cousin rather than sister cell fusions, further contributing to pathological genomic rearrangements (Fig. 4e). To quantify the impact of defective mitotic exit on subsequent cell cycle progression, we employed a custom cell tracking software and analysed cell lineage development in C-604-treated HCC1395, RPE-1 and HeLa cells. In all three cell line models, exposure to increasing doses of C-604 resulted in a gradual decrease in the number of cell divisions per lineage (Fig. 4f, g). In agreement with our preliminary assessment, mitotic failure in HCC1395 and RPE-1 cells

appeared to suppress further mitotic commitments (Fig. 4f, g, h). Notably, some C-604-treated HCC1395 and RPE-1 cells did not divide during the entire course of imaging (Fig. 4f). These cells showed only subtle signs of mitotic entry, including cell rounding as well as maturation and separation of centrosomes, followed by reversal into the interphase (Supplementary Fig. 10). In stark contrast to HCC1395 and RPE-1 cells, HeLa cells maintained proliferative capacity and underwent a series of unsuccessful divisions and cellular fusions (Fig. 4f, g, h). In agreement with our analysis of long-term C-604 effects, this sequence of futile cell cycles led to pronounced polyploidisation and deterioration of nuclear architecture (Fig. 2 and

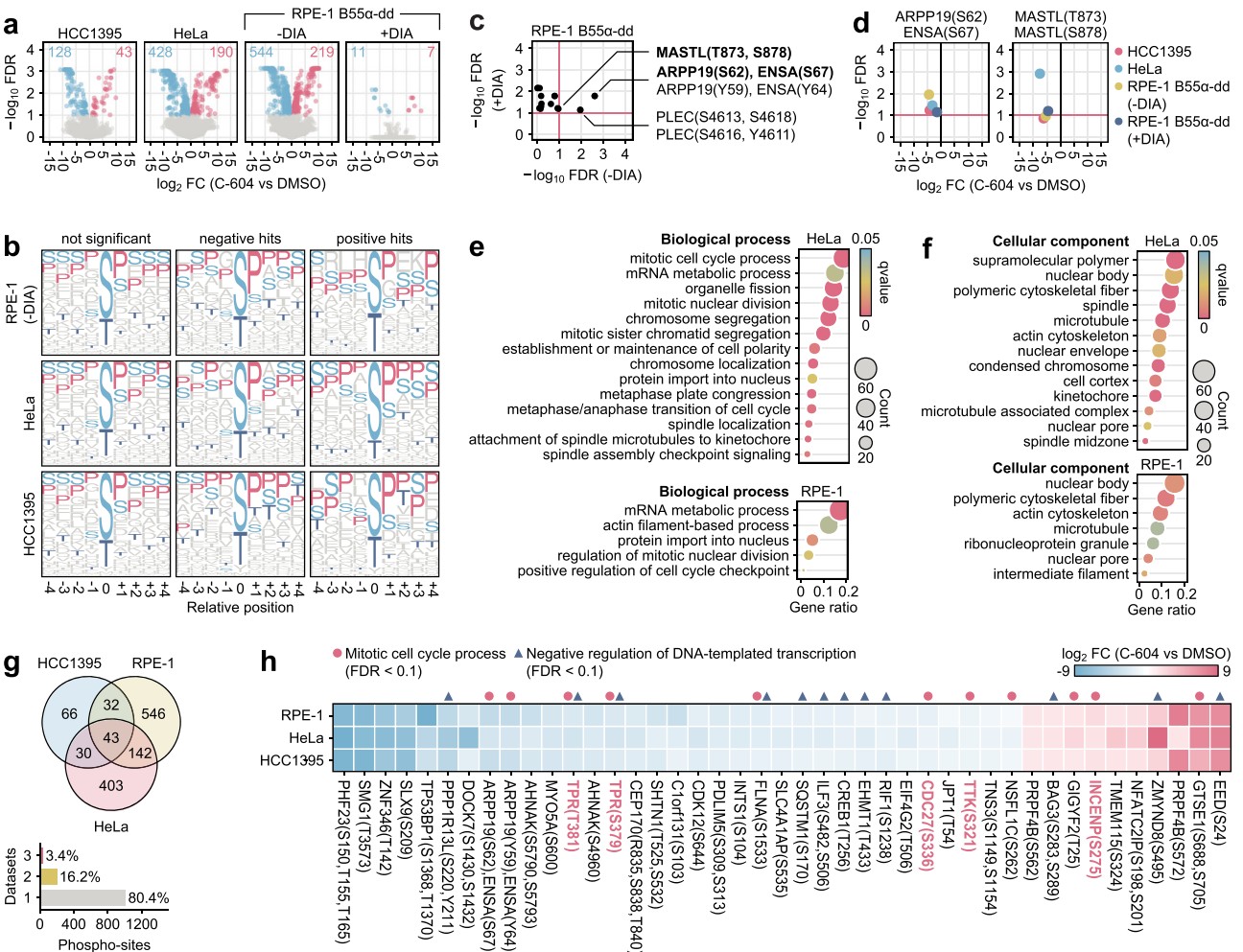

**Fig. 5 | Characterisation of phosphoproteomic changes induced by C-604-mediated GWL inhibition. a** Volcano plots displaying changes in phosphoproteomes of prometaphase-arrested cells treated with 2 μM C-604 for 30 min. False discovery rate (FDR) was calculated from *n* = 3 independent repeats. An FDR cut-off (FDR ≥ 0.1) was used as a statistical significance threshold. FC – fold change. **b** Sequence motifs of identified phosphosites with unchanged (not significant) and significantly impacted (negative and positive hits) phosphorylation levels. **c** FDR values of de-phosphorylated phosphosites identified in B55α-depleted RPE-1 B55α-dd ( + DIA) cells against FDR values of respective phosphosites in control RPE-1 B55α-dd (-DIA) cells. **d** FC and FDR values of ARPP19(S62), ENSA(S67) and MASTL

(T873, S878) phosphosites in indicated cell lines. **e, f** Biological process (e) and cellular component (f) gene ontology terms significantly enriched in phosphopeptides dephosphorylated in prometaphase-arrested RPE-1 B55α-dd (-DIA) and HeLa cells treated with 2 μM C-604. **g** Intersection analysis of differentially phosphorylated phosphosites in prometaphase HCC1395, RPE-1 B55α-dd (-DIA) and HeLa cells treated with 2 μM C-604. **h** FC values of 43 differentially phosphorylated phosphosites identified in all three analysed cell lines, including HCC1395, RPE-1 B55α-dd (-DIA) and HeLa. The phosphosites highlighted in red indicate factors involved in metaphase-anaphase progression.

Fig. 4a). Overall, the live-cell imaging analysis established the premature mitotic exit and defective cytokinesis as the root causes of pleiotropic pathologies associated with C-604-mediated GWL inhibition. The cell-tracking-based assessment of cell lineage development also demonstrated that the extensive polyploidisation observed in HeLa and BT-549 cells arose from the inability to stall cell cycle progression following the defective mitotic resolution.

**Inhibition of GWL destabilises the mitotic phosphoproteome**

To determine the molecular foundations of cellular pathologies associated with C-604-mediated GWL inhibition, we analysed the phosphoproteomes of prometaphase-arrested HeLa, HCC1395 and RPE-1 cells treated with C-604. To distinguish between the changes in phosphorylation levels of direct GWL substrates and the effects of de-inhibited PP2A-B55α phosphatase, we utilised RPE-1 Tet-OsTIR1 B55α-mAID-SMASh cells (RPE-1 B55α-dd), which degrade B55α when treated with doxycycline, indole-3-acetic acid and Asunaprevir (DIA) (Supplementary Fig. 11a). We tagged B55α with two complementary degron

motifs, mAID[37] and SMASh[38], as the combination of these degrons improves degradation efficiency and minimises the residual levels of targeted protein[39]. A 30-minute exposure to 2 μM C-604 led to a substantial destabilisation of mitotic phosphoproteomes, resulting in the dephosphorylation of 128, 428, and 544 phosphosites, along with increased phosphorylation at 43, 190, and 219 phosphosites in HCC1395, HeLa, and RPE-1 B55α-dd (-DIA) cells, respectively (Fig. 5a). Identified differentially phosphorylated peptides displayed a pronounced sequence bias for proline at the +1 position, a well-characterised minimum consensus motif of Cdk1 substrates[40,41], suggesting that C-604-mediated GWL inhibition predominantly impacted mitotic factors regulated by the Cdk1 kinase (Fig. 5b). The significant differences in the overall impact of C-604 indicated differential stability of mitotic phosphoproteomes, further reinforcing the notion of cell-specific dependencies on GWL activity (Fig. 5a).

Importantly, DIA-induced degradation of B55α resulted in a near-complete rescue of phosphoproteomic changes induced by C-604 in RPE-1 B55α-dd ( + DIA) cells (Fig. 5a). This result was consistent with the

observation that cytotoxic properties of C-604 depended on B55α (Figs. 1i, j, 3, Supplementary Fig. 6 and Supplementary Fig. 9d–g), providing further evidence that PP2A-B55α phosphatases represent the primary drivers of physiological defects associated with GWL inactivation. Following the administration of C-604, B55α-depleted RPE-1 cells displayed only 11 dephosphorylated and 7 hyperphosphorylated phosphosites (Fig. 5a, Supplementary Fig. 11b). Among the 11 dephosphorylated peptides, which represented potential direct GWL substrates, only two serine/threonine sites showed significant changes in control RPE-1 B55α-dd (-DIA) cells (Fig. 5c). These included the canonical GWL substrate ARPP19(S62), ENSA(S67) and the cytoskeleton interlinking protein PLEC(S4613, S4618) (Fig. 5c). The C-604-induced dephosphorylation of the ARPP19(S62), ENSA(S67) site was consistent across all tested cell lines (Fig. 5d). In contrast, phosphorylation levels of the identified PLEC phosphosite did not exhibit significant changes in HCC1395 and HeLa cells following the administration of C-604, indicating that PLEC(S4613, S4618) did not represent a genuine GWL substrate (Supplementary Fig. 11c). C-604-treated RPE-1 cells lacking B55α also exhibited significant dephosphorylation of the GWL C-terminus, MASTL(T873, S878), previously characterised as the auto-phosphorylation site of GWL[30,42,43] (Fig. 5c and Supplementary Fig. 11b). However, due to the limited distance between possible C-terminal GWL phosphosites (T873, S875, S878), we could not determine the exact phosphoresidue(s) affected by C-604. The C-604-induced reduction in phosphorylation of MASTL(T873, S878) was also evident in HeLa cells, and only narrowly failed to meet the stringent false discovery rate threshold in control RPE-1 B55α-dd (-DIA; FDR = 0.11 > 0.1; $p = 0.005$) and HCC1395 (FDR = 0.14 > 0.1; $p = 0.0018$) cells (Fig. 5d). In conclusion, ARPP19(S62), ENSA(S67), and the GWL auto-phosphorylation site MASTL(T873, S878) were the only sites consistently dephosphorylated in a PP2A-B55α-independent manner and represented the only GWL substrates identified with statistically stringent confidence.

Differentially phosphorylated proteins identified in HeLa cells displayed significant enrichment in processes related to mRNA metabolism and chromosome segregation (Fig. 5e). These peptides were associated with cellular components, including mitotic spindle, actin cytoskeleton, condensed chromatin, and nuclear envelope (Fig. 5f). In RPE-1 B55α-dd (-DIA) cells, C-604-induced dephosphorylation impacted proteins involved in mRNA metabolism, nuclear division, actin cytoskeleton organisation, and nuclear transport (Fig. 5e). C-604-impacted factors specific to RPE-1 B55α-dd (-DIA) cells were associated with intracellular structures such as cytoskeletal networks, nuclear pores, and ribonucleoprotein complexes (Fig. 5f). Under stringent filtering conditions (adjusted $p < 0.05$, FDR < 0.1), differentially phosphorylated peptides identified in HCC1395 cells did not demonstrate any significant gene ontology (GO) enrichment. Overall, C-604-mediated GWL inhibition affected diverse processes fundamental to unperturbed mitotic progression.

Strikingly, only 3.4% ($n = 43$) of all identified phosphosites significantly affected by C-604 appeared consistently across all three cell lines HCC1395, HeLa and RPE-1 B55α-dd (-DIA) (Fig. 5g). These 43 hits included a broad range of proteins enriched in factors involved in the mitotic cell cycle process ($n = 10$) and negative regulation of transcription ($n = 11$) (Fig. 5h). Within the group of cell cycle proteins, we identified factors essential for chromosome segregation and metaphase-anaphase progression, including significantly dephosphorylated nuclear basket protein TPR(S379 and T381), the APC/C subunit CDC27(S336) and the dual specificity protein kinase TTK(S321), together with the hyper-phosphorylated chromosome passenger complex (CPC) subunit INCENP(S275) (Fig. 5h). Other independent mass spectrometry studies have identified TPR(S379), CDC27(S336), TTK(S321) and INCENP (S275) as genuine mitotic substrates[44–47]; however, to our best knowledge, the functional relevance of these phosphosites remains to be determined.

Collectively, our attempts to characterise the systemic impact of GWL inhibition on mitotic phosphorylation highlighted ARPP19(S62) and ENSA(S67) as the only direct GWL substrates and strongly suggested that C-604-induced destabilisation of the mitotic phosphoproteome represented a direct consequence of untimely PP2A-B55α activation. Notably, the impact of C-604 on mitotic phosphorylation showed significant qualitative and quantitative differences between distinct cell line models, highlighting the complex nature of mitotic signalling networks and their regulation. Finally, GO enrichment and comparative intersection analyses indicated that C-604-induced mitotic defects originated from pleiotropic deregulation of chromosome segregation and metaphase-anaphase transition.

### Cross-referencing of putative PP2A-B55α substrates

Since phosphoproteomic changes and cellular pathologies induced by C-604 strictly depended on PP2A-B55α, we presumed that many detected dephosphorylated peptides represented PP2A-B55α substrates. To cross-validate our findings, we examined the overlap between phosphosites negatively affected by C-604 and three independent datasets of experimentally determined PP2A-B55 substrates[48–50]. These publicly available lists of putative PP2A-B55 substrates showed moderate overlaps with each other (up to approximately 23% between a pair of datasets); however, they encompassed relatively few C-604-responsive phosphosites (Supplementary Fig. 12a, b). The most prominently represented C-604-sensitive phosphosites included the known PP2A-B55 substrates ENSA(S67) and ARPP19(S62), as well as phosphosites with hypothetical roles in chromosome segregation and metaphase-anaphase progression, such as TTK(S321), TPR(S379) and TOP2B(S1236) (Supplementary Fig. 12c). Additional well-represented phosphosites included SMTN(S357), SURF6(T229), RIF1(S1238), the RPE-1-specific LASP1(T104), SMTN(S341) and SMTN(S729), as well as the HeLa-specific NUMA1(T2000) (Supplementary Fig. 12c). We believe the significant discrepancy between our study and other analysed datasets originates from fundamental differences in the employed experimental designs and models used in these studies. While our experiments measured changes in phosphorylation levels during prometaphase following a brief (~30-minute) inhibition of GWL, the PP2A-B55 substrates identified by Cundell et al. were inferred from nocodazole-arrested HeLa cells depleted of all B55 subunits (B55α–δ) or GWL for 72 h before sample collection[48]. We argue that the cellular consequences of acute and long-term suppression of GWL function are likely distinct and may not be directly comparable. Similarly, studies by Kruse et al. and Hein et al. identified PP2A-B55 substrates using mitotic protein extracts[49,50], which may differ from those detected in intact cells.

### GWL and B55α expression levels predict sensitivity to GWL inhibition

Our efforts to characterise cellular consequences of GWL inhibition strongly suggested that the requirement for unperturbed GWL activity varies across distinct experimental models. To uncover mechanisms underlying differences in cellular GWL dependencies, we quantified the impact of GWL inhibition on colony formation in a larger experimental panel composed of 12 distinct cell lines (Fig. 6a). The analysed cell line panel displayed a broad range of C-604 median effective doses (ED$_{50}$), ranging from 0.09 ± 0.04 μM in the most sensitive BT-549 cells to 2.03 ± 0.48 μM in the most resistant HCC1143 cell line (Fig. 6b). Looking for the simplest plausible explanation, we speculated that observed differences reflected varying intracellular levels of GWL, PP2A-B55α and, possibly, ENSA/ARPP19. To test this hypothesis, we correlated established C-604 ED$_{50}$ values with protein expression levels of GWL, B55α, PP2A catalytic subunit isoforms α and β (PPP2Cα/β) and ENSA/ARPP19 (Fig. 6c, d). Strikingly, cellular sensitivity to GWL inhibition (represented by established C-604 ED$_{50}$ values) strongly

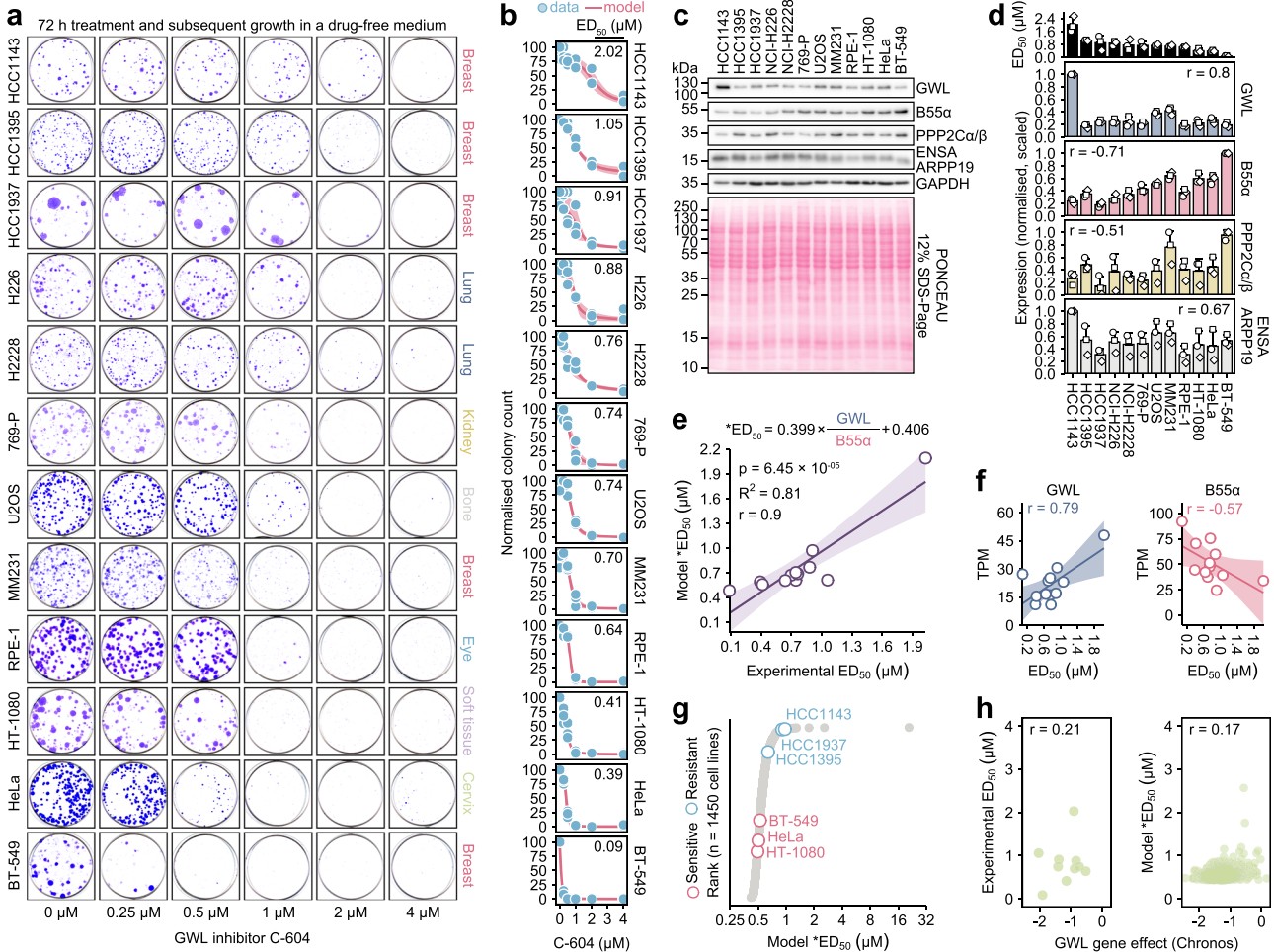

**Fig. 6 | Prediction of cellular sensitivity to GWL inhibition. a** Representative images from one of $n = 3$ independent experiments demonstrating colony formation capacity of indicated cell lines exposed to C-604 for 72 h and allowed to recover for 7-14 days. The histological origins of the tested cell lines are indicated. **b** Quantification of colony formation following the exposure to increasing doses of C-604 (a). Dots represent the results of three independent experiments. Red lines and shaded regions represent means and standard deviations of three independently fitted four-parameter sigmoidal models. Mean $ED_{50}$ values are indicated. **c** Representative Western blot showing cellular protein levels of GWL, B55α, catalytic PP2A subunit PPP2Cα/β and ENSA/ARPP19. GAPDH was used as a loading control. Scans from one of three experiments are shown. **d** Quantification of indicated protein levels aligned with respective C-604 $ED_{50}$ values. In each of $n = 3$ experiments, protein levels were normalised to GAPDH and scaled so that the highest value equalled 1. Bars and error bars represent the means and standard deviations of three repeats. Pearson correlation coefficients of C-604 $ED_{50}$ values and protein expression levels are indicated. **e** Correlation scatterplot

demonstrating mean experimental C-604 $ED_{50}$ and predicted $*ED_{50}$ values. The mathematical model used to predict $*ED_{50}$ values is indicated. In the equation, GWL and B55α represent normalised, scaled protein expression levels, as shown in (d). p-value, $R^2$ value and Pearson correlation coefficient are indicated. Statistical significance was determined using a two-tailed t-test as part of the function that calculated the Pearson correlation. The shaded region represents the confidence interval. **f** Correlations of experimental C-604 $ED_{50}$ and publicly available RNA expression levels obtained from the DepMap repository[51]. RNA expressions are represented as transcript per million (TPM) values. Pearson correlation coefficients are indicated. Shaded regions represent confidence intervals. **g** Stratification of 1450 cell lines based on TPM-derived C-604 $*ED_{50}$ predictions. The most sensitive (BT-549, HeLa, HT-1080) and the most resistant (HCC1395, HCC1937, HCC1143) cell lines, as determined experimentally, are indicated. **h** Correlation scatterplots of experimental C-604 $ED_{50}$ or predicted $*ED_{50}$ values and GWL CRISPR gene dependency scores obtained from the DepMap repository[51,69]. Pearson correlation coefficients are indicated.

correlated with GWL and B55α protein levels (Fig. 6d). Established C-604 $ED_{50}$ values also correlated with expressions of PPP2Cα/β and ENSA/ARPP19, albeit to a lesser extent (Fig. 6d).

We employed linear modelling without weighting and ordinary least squares regression to establish a mathematical model predicting cellular sensitivity to GWL inhibition. Our model was based on the modified ratio of GWL and B55α protein levels (Model $*ED_{50} = 0.399 \times$ GWL / B55α + 0.406) and accurately predicted experimental C-604 $ED_{50}$ values ($R^2 = 0.81$, $p = 6.45 \times 10^{-05}$) (Fig. 6e). In support of the utility of GWL and B55α expression patterns as versatile biomarkers of GWL dependency, we also found a correlation between experimentally determined C-604 $ED_{50}$ values and GWL and B55α RNA expression levels derived from the publicly available RNA-Seq dataset[51] (Fig. 6f). Indeed, when applied to the publicly available RNA expression data of

1450 cell lines[51], our model successfully segregated the most sensitive (BT-549, HeLa and HT-1080) and the most resistant (HCC1395, HCC1937 and HCC1143) cell lines (Fig. 6g). Notably, neither experimental nor predicted $ED_{50}$ values correlated with publicly available GWL CRISPR dependencies[51] (Fig. 6h). We argue that variability in sensitivities to chemical suppression of GWL activity mainly reflects differences in cellular responses to partial GWL inhibition by lower C-604 concentrations (Fig. 6a, b). CRISPR-driven GWL depletion, however, only captures cellular responses to near-complete GWL inactivation, failing to account for the effects of a partial reduction in GWL activity.

To evaluate the predictive capacity of our model in cellular systems not used during its development, we correlated the predicted and experimentally determined sensitivities to C-604 treatment across

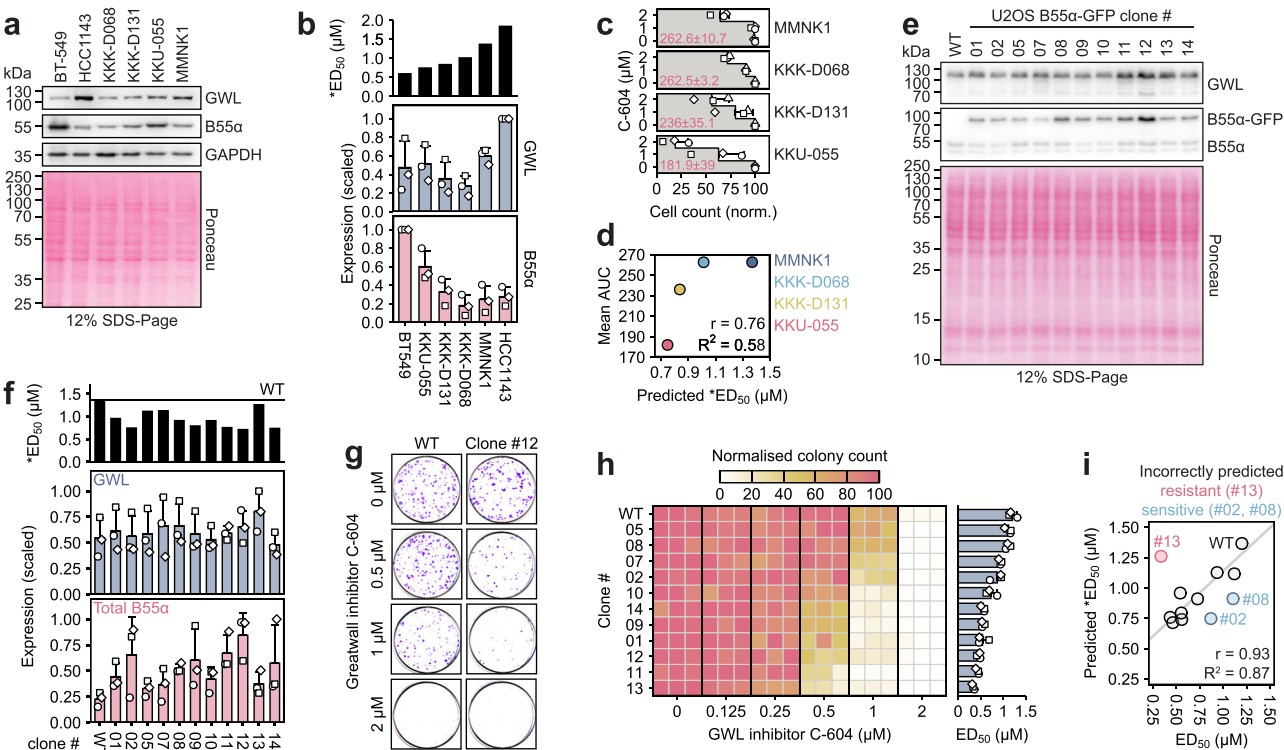

**Fig. 7 | Modulation of GWL dependency by genetic manipulation of B55α expression levels. a** Representative Western blot showing GWL and B55α protein expression levels in CHOL cancer cell lines KKK-D068, KKK-D131, KKU-055 and MMNK1. Scans from one of $n = 3$ independent experiments are shown. BT-549 and HCC1143 cell lines are included as exemplars of high and low cellular C-604 sensitivity, respectively. **b** Quantification of $n = 3$ independent measurements of GWL and B55α protein expression levels accompanied by predicted C-604 *ED_{50} values. In each experiment, the recorded expression levels were scaled, with the maximum value set to 1. Points represent results of $n = 3$ independent experiments (circle, square, rhombus). Bars and error bars represent means and standard deviations. **c** The impact of C-604 on proliferation in indicated cell lines. Cell counts were determined by microscopy and normalised to the maximum values. Bars and error bars represent means and standard deviations of $n = 3$ independent experiments, respectively. Mean AUC values computed from normalised counts ± standard deviations are indicated. AUC – area under the curve. **d** Correlation scatterplot of predicted *ED_{50} and mean AUC values (**b**, **c**) in indicated CHOL cancer cell lines (**e**) Representative Western blot showing GWL and B55α expression levels in WT and B55α-overexpressing U2OS cells. Scans from one of $n = 3$ experiments are shown. **f** Quantification of GWL and total B55α protein levels accompanied by predicted

C-604 *ED_{50} values in indicated cell lines. Points represent results of $n = 3$ independent experiments (circle, square, rhombus). Bars and error bars represent means and standard deviations, respectively. **g** Representative result of the colony formation assay showing WT and B55α-overexpressing (clone #12) U2OS cell lines exposed to C-604 for 72 h. Images from one of $n = 3$ independent experiments are shown. **h** Heatmap showing colony formation capacity of WT and B55α-overexpressing U2OS cells treated with C-604 for 72 h and allowed to recover for 10 days. Results of $n = 3$ independent experiments are shown as separate columns. Bars and error bars in the bar plot represent the means and standard deviations of C-604 ED_{50} estimates. Points represent results of $n = 3$ independent experiments (circle, square, rhombus). **i** Correlation scatterplot of experimental C-604 ED_{50} and predicted *ED_{50} values in WT and B55α-overexpressing U2OS cell lines. The incorrectly predicted clones are indicated. The solid line represents the linear model based on correctly predicted sensitivities to C-604 treatment. r – Pearson correlation coefficient, $R^2$ – coefficient of determination. Predicted C-604 sensitivity was determined using the formula *ED_{50} = 0.399 × GWL / B55α + 0.406, where GWL and B55α represent scaled GWL and B55α protein expression levels, respectively.

four cholangiocarcinoma (CHOL) cell lines, including KKK-D068, KKK-D131, KKU-055, and MMNK1[52]. Using B55α and GWL expression levels, we predicted a range of *ED_{50} values, with KKU-055 and MMNK1 exhibiting the highest and lowest predicted sensitivity to C-604, respectively (Fig. 7a, b). To experimentally assess cellular sensitivity to GWL inhibition, we quantified the effect of C-604 on the proliferation of control and C-604-treated cell cultures (Fig. 7c). Proliferation defects induced by C-604 treatment correlated with the predicted *ED_{50} values, demonstrating the reliability of the model's predictions (Fig. 7d).

To further scrutinise and validate the model's predictive power, we utilised a random plasmid integration technique and generated 11 clonal U2OS cell lines expressing different levels of B55α-GFP (Fig. 7e, f). Differential upregulation of B55α was not associated with significant changes in GWL expression levels and thus represented the primary driver of predicted C-604 sensitivities (Fig. 7f). To determine whether model C-604 *ED_{50} values predicted the cellular responses to GWL inhibition, we assessed the colony formation capacity in all eleven

B55α-overexpressing cell lines exposed to increasing doses of C-604 (Fig. 7g, h). Our model accurately predicted cellular sensitivity to C-604-mediated GWL inhibition in WT U2OS cells and 8 out of 11 B55α-overexpressing clones, further reinforcing the potential of GWL and B55α expression levels as robust predictors of cellular GWL dependency (Fig. 7i). The discrepancy between predicted and experimental GWL dependencies in 3 out of 11 B55α-overexpressing clones suggested that other factors may shape cellular sensitivity to GWL inhibition. In summary, while the involvement of additional predictors cannot be excluded, our analysis identified B55α and GWL expression levels as key determinants of differential cellular sensitivity to chemical inhibition of GWL activity.

**TCGA analysis highlights rare cases with potential sensitivity to GWL inhibition**

To evaluate whether the postulated prediction of sensitivity to GWL inhibition has any clinical significance, we assessed RNA expression profiles of GWL and B55α in biopsies from normal and cancer tissues

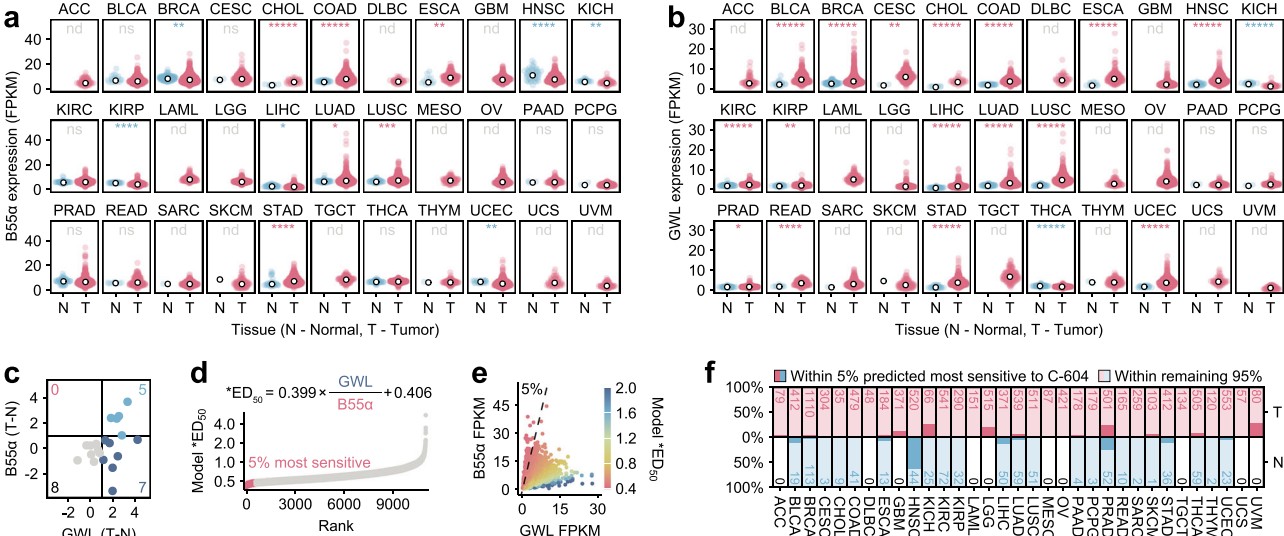

**Fig. 8 | Profiling of B55α and GWL RNA expression levels in TCGA cancer-derived biopsies. a, b** RNA expression levels of B55α (a) and GWL (b) in 33 distinct cancer groups. Red and blue points represent expression levels derived from independent matching normal (N) and Tumour (T) tissue samples. White dots represent median values. The number of analysed biopsies is shown in (d). Statistical significance was determined using an unpaired two-tailed Wilcoxon test. Determined p-values were adjusted for multiple testing using the Benjamini-Hochberg procedure. Indicated significance is colour-coded to reflect the positive (red) and negative (blue) differences between T and N samples. nd – not determined, ns – not significant, * $p < 0.05$, ** $p < 0.01$, *** $p < 0.001$, **** $p < 0.0001$, ***** $p < 0.00001$. FPKM – fragments per kilobase of transcript per million mapped reads. **c** Scatter plot displaying differences in median B55α and GWL RNA expression levels (a, b) between tumour and matching normal tissues (T-N). **d** Ranked predictions of C-604 sensitivity (*ED$_{50}$) based on B55α and GWL RNA expression levels (a, b) in analysed biopsies. *ED$_{50}$ values were calculated using the indicated formula. 5% of biopsies predicted to be most sensitive to C-604 treatment (the lowest *ED$_{50}$) are indicated. **e** Scatterplot representation of predicted *ED$_{50}$ values.

5% of the samples with the lowest *ED$_{50}$ are indicated. **f** Proportions biopsies with the lowest predicted *ED$_{50}$ values (5%) in distinct groups of tumours and matching normal tissues. Total cases per group are indicated. ACC adrenocortical carcinoma, BLCA bladder urothelial carcinoma, BRCA breast invasive carcinoma, CESC cervical squamous cell carcinoma and endocervical adenocarcinoma, CHOL cholangiocarcinoma, COAD colon adenocarcinoma, DLBC lymphoid neoplasm diffuse Large B-cell lymphoma, ESCA oesophageal carcinoma, GBM glioblastoma multiforme, HNSC head and neck squamous cell carcinoma, KICH kidney chromophobe, KIRC kidney renal clear cell carcinoma, KIRP kidney renal papillary cell carcinoma, LAML acute myeloid leukaemia, LGG brain lower grade glioma, LIHC liver hepatocellular carcinoma, LUAD Lung adenocarcinoma, LUSC Lung squamous cell carcinoma, MESO mesothelioma, OV ovarian serous cystadenocarcinoma, PAAD pancreatic adenocarcinoma, PCPG pheochromocytoma and paraganglioma, PRAD prostate adenocarcinoma, READ rectum adenocarcinoma, SARC sarcoma, SKCM skin cutaneous melanoma, STAD stomach adenocarcinoma, TGCT testicular germ cell tumours, THCA thyroid carcinoma, THYM thymoma, UCEC uterine corpus endometrial carcinoma, UCS uterine carcinosarcoma, UVM uveal melanoma.

obtained from the Cancer Genome Atlas (TCGA) repository, generated by the TCGA Research Network (Fig. 8a, b). First, we examined the differences in median B55α and GWL expression levels in tumours and matched normal tissues. Strikingly, the combination of elevated B55α and reduced GWL expression levels, indicative of high sensitivity to GWL inhibition, was not observed in any of the 20 analysed cancer groups (Fig. 8c). This result suggested that B55α overexpression, concomitant with GWL downregulation, represents a systemic feature selected against during tumour development. Next, we used GWL and B55α RNA expression levels to predict susceptibility to GWL inhibition in individual samples. Most biopsies exhibited moderate or low predicted sensitivity to GWL inhibition, with only a marginal increase in 5% of cases, characterised by the highest projected sensitivity scores (Fig. 8d, e). Notably, the biopsies predicted to be most sensitive to GWL inhibition were overrepresented in kidney chromophobes (KICH) and, possibly, in low-grade glioma (LGG) and uveal melanoma (UVM) cancers (Fig. 8f). However, due to the lack of matched normal tissue samples, we could not validate whether the over-representation of samples sensitive to GWL inhibition (by prediction) in LGG and UVM represented a relevant observation (Fig. 8f). Collectively, the analysis of TCGA expression data strongly indicated that the hyper-sensitivity to GWL inhibition driven by high B55α and low GWL expression levels represents a rare biological occurrence. However, since B55α and GWL are unlikely to be the only determinants of cellular sensitivity to GWL inhibition, further studies are required to determine the true clinical potential of chemical GWL inhibition.

## Discussion

The key mitotic regulator, GWL kinase, has received increasing attention as a putative oncogene and prospective chemotherapeutic target; however, our understanding of the effects of chemical GWL inactivation on fundamental cellular processes remains incomplete. This work presents a detailed classification of cellular and molecular responses to chemical suppression of GWL activity and proposes predictive biomarkers of cellular GWL dependency.

To characterise cellular phenotypes associated with chemical GWL inactivation, we developed and validated a tool compound, C-604, which interferes with GWL activity in prometaphase-arrested human cells and *Xenopus laevis* egg extracts. We identified the potential C-604 off-targets and demonstrated that the effects of C-604 stem from inhibition of GWL rather than other kinases. Our characterisation of the alternative GWL inhibitor MKI-2 demonstrated that MKI-2 exhibits strong off-target activity, rendering C-604 the more suitable GWL-targeting tool compound with sufficient biological characterisation.

Using live-cell imaging and high-throughput immunofluorescence microscopy, we determined that GWL inhibition triggers premature mitotic exit followed by an incomplete separation or fusion of the newborn daughter cells. The observed mitotic defects were highly reminiscent of the phenotypes associated with genetic GWL depletion[12–17], indicating that genetic and chemical means of GWL inhibition result in equivalent cellular consequences. Cell division defects consequential to reduced GWL activity did not cause

immediate cell death, but rather led to polyploidisation, nuclear fragmentation and a gradual increase in cell size. Notably, the impact of C-604-induced mitotic exit defects on subsequent cell cycle progression varied in different cell line models. For instance, HeLa and BT-549 cells retained proliferative potential and underwent additional cycles of defective cell divisions and fusions, leading to further physiological pathologies and pronounced genomic amplification. The long-term C-604-induced phenotypes were considerably different in untransformed RPE-1 cells, which upregulated the cell cycle inhibitor p21, suppressed DNA replication and entered a cell cycle arrest with 2 N or 4 N DNA content. Regardless of the long-term phenotypic outcome, C-604-induced mitotic exit defects prevented normal cellular proliferation in all analysed cell line models, strongly implying that mitotic failure represents the root cause of the cytostatic effects accompanying GWL inhibition.

Interestingly, the fundamental differences in long-term phenotypic responses to C-604 treatment did not reflect cellular p53 status and the ability to upregulate p21, suggesting the involvement of p53- and p21-independent cell cycle control. While this remains speculative, this regulation may involve the p53 family member p73, which has been shown to suppress aneuploidy and polyploidy in p53-deficient mouse embryonic fibroblasts through a p21-independent mechanism[53,54]. Although potential p53-independent cell cycle control responding to signs of genomic instability is a highly relevant subject, investigation of this phenomenon did not fall within the scope of this manuscript and will be further explored in future studies.

We present substantial evidence that cellular pathologies associated with GWL inhibition stem from the cytotoxic activation of PP2A-B55α phosphatases. The depletion of B55α by siRNAs decreased the cellular dependency on GWL activity, resulting in the near-complete rescue of C-604-induced pathologies and proliferation defects. This result strongly indicated that C-604 acted on-target and interfered with the PP2A-B55/ENSA/GWL regulatory axis. Notably, cellular responses to C-604 treatment were unaffected by the depletion of other B55 isoforms (B55β, B55γ and B55δ), implying that PP2A-B55α complexes are the primary drivers of cellular susceptibility to GWL inhibition. In agreement with this assessment, an in-depth analysis of mitotic phosphoproteomes revealed that PP2A-B55α phosphatases are responsible for the majority of phosphoproteomic changes accompanying the acute loss of GWL activity. Importantly, in support of the primary mitotic role of GWL as a PP2A-B55 suppressor, we identified GWL autophosphorylation site, MASTL(T873, S878), and PP2A-B55 inhibitors ENSA(S67) and ARPP19(S62) as the only high-confidence mitotic GWL substrates. Consequently, we argue that the cytotoxic potential of GWL inhibition manifests through the dephosphorylation of canonical GWL substrates ENSA(S67) and ARPP19(S62) and the resulting de-inhibition of PP2A-B55α. We hypothesise that the untimely increase in PP2A-B55α activity triggers futile phosphorylation-dephosphorylation cycles, which destabilise the phosphorylation of Cdk1 substrates, functionally deteriorate cell division machinery and ultimately result in a premature mitotic exit with deleterious consequences.

Our assessment of cellular responses to C-604 treatment reveals that sensitivity to GWL inhibition varies across distinct cell line models. We identified cell-specific variations in C-604-induced proliferation defects and significant inconsistencies in the frequency of aberrant mitotic events. Additionally, we documented quantitatively and qualitatively distinct impacts of acute GWL inhibition on mitotic phosphorylation in prometaphase-arrested HCC1395, RPE-1 and HeLa cells. We demonstrated that these cell-specific discrepancies reflect differences in B55α and GWL expression levels and postulated a mathematical model predicting cellular sensitivity to GWL inhibition. We propose that a high abundance of intracellular B55α enhances the cytotoxic potential of unscheduled mitotic PP2A-B55α activation and increases cellular dependency on the PP2A-B55-counteracting GWL

activity. This model aligns with the fundamental principles of the GWL/PP2A/Cdk1 signalling circuit and highlights B55α-incorporating PP2A complexes as determinants of cellular sensitivity to GWL-targeting compounds with potential utility in oncological studies.

The functional and clinical significance of B55α in tumour development is currently unclear, with contradicting evidence suggesting oncogenic properties in pancreatic malignancies[55] and tumour-suppressing roles in prostate, ovarian, and breast cancers[56–59]. In addition to these roles, we propose that B55α determines cellular dependency on GWL activity and drives cellular susceptibility to GWL inhibition. According to our model, tissues with high cellular abundance of B55α and reduced GWL expression levels do not tolerate further loss of GWL activity and represent prospective targets of GWL-focused therapy.

Although the utility of GWL inhibitors in oncology remains to be determined, our in-depth analysis of TCGA cancer expression profiles suggests that simultaneous upregulation of B55α and downregulation of GWL is relatively rare. Consequently, the application of GWL-targeting therapeutics might be limited to a small group of tumours with uncommon genetic architecture. However, further research is required to investigate the existence of additional predictive markers and combinatorial vulnerabilities, which could enhance our understanding of the actual clinical potential of GWL-targeted interventions. Ultimately, broader development and detailed characterisation of GWL-targeting compounds will be invaluable in identifying additional biological roles of GWL and their functional significance in diverse physiological and disease-related contexts.

## Methods

### Homogeneous time-resolved fluorescence screening
4 μL of the assay buffer (30 mM HEPES, 100 mM NaCl, 0.5 mM EGTA, 10 mM MgCl$_2$, 0.01% Tween-20, 0.5 mM TCEP added at the time of use) were introduced into each well of 384-well white assay plates. Assay-ready plates contained 100 nL of a 1 mM compound, resulting in a final screening concentration of 10 μM in a total volume of 10 μL. Alternatively, 384-well plates were loaded with 4 μL from an intermediate compound dilution plate. Subsequently, 2 μL of enzyme solution (770 pM) or the control assay buffer was introduced. The plate was gently mixed, briefly centrifuged (1000 × g, 1 min) and incubated at room temperature for 15 min. Reactions were completed by adding 4 μL of the ATP/ENSA peptide substrate mix (25 nM ENSA, 45 μM ATP in). The plate was gently mixed, briefly centrifuged (1000 × g, 1 min), sealed and incubated at 37 °C for 60 min. The reactions were mixed with 10 μL Ab-K/SA-D2, gently mixed, centrifuged (1000 × g, 3 min) and incubated at room temperature for 60 min. Fluorescence intensity values were assessed with the PHERAstar Microplate Reader (BMG).

### Determination of potential off-target kinases
The in vitro off-target effects of compound C-604 were determined by assessing its inhibitory potential in a panel of 100 human kinases using the LanthaScreen Kinase Activity Assay (Thermo Fisher Scientific).

### GWL inhibition in CSF extract
Compound C-604 was added to CSF extract at 23 °C, with DMSO as a negative control. Purified 10His-Ensa (1 μM) was added to each tube to start the reaction. Interphase extracts were prepared by adding 0.1 μg/μL cycloheximide and 0.4 mM CaCl$_2$ to CSF extract at 23 °C for 1.5 h. Samples collected at the indicated time points were mixed with sample buffer and boiled. Phos-tag gels contained 20% acrylamide, Phos-tag solution (30 μM) and MnCl$_2$ (30 μM).

### Human cell culture
Cells were incubated at 37 °C, 95% humidity and 5% CO$_2$. hTERT RPE-1, HeLa, U2OS, MDA-MB-231, KKK-D068, KKK-D131, KKU-055, and MMNK1 cells were grown in Dulbecco's modified Eagle medium,

DMEM (Gibco, 21969035). HCC1143, HCC1395, HCC1937, BT-549, 769-P, NCI-H226 and NCI-H2228 cells lines were grown in the RPMI-1640 medium (Gibco, 31870025). HT-1080 cells were grown in the minimum essential medium, MEM (Gibco, 21090022), supplemented with non-essential amino acids (Gibco, 11140050). For live-cell imaging, cells were cultured in FluoroBrite DMEM (Gibco, A1896701). Media were standardly supplemented with 2 mM L-glutamine (Sigma-Aldrich, G7513), penicillin-streptomycin (Sigma-Aldrich, P4333) and 10% fetal calf serum (Gibco, A5670701). DMEM medium used to grow KKK-D068, KKK-D131 and MMNK1 cells was supplemented with 2 mM L-glutamine (Sigma-Aldrich, G7513), penicillin-streptomycin (Sigma-Aldrich, P4333) and 5% fetal calf serum (Gibco, A5670701). Cells were passaged every 2-5 days, depending on the cell line. Cells were cultured for up to three weeks.

### Generation of a stable cell line using the lentiviral transduction
The following procedure was used to generate stable cell lines expressing α-tubulin-GFP and H2B-mCherry. Lentiviral packaging was done in HEK293T cells. Plasmids were transfected using the calcium/phosphate mammalian transfection kit (Clontech Laboratories, 361312). The virus was assembled from pCMV-VSV-G (Addgene, 8454) and psPAX2 (Addgene, 12260) and packaged with plasmids pLenti6-H2B-mCherry (Addgene, 89766) and L304-EGFP-Tubulin-WT (Addgene, 64060). 100 μL of the transfection mixture (HEPES-buffered saline, 2.5 μg plasmid DNA, 124 mM calcium phosphate) was incubated at room temperature for 10 min and added dropwise onto HEK293T cells (50-80% confluency). Transfected cells were subsequently incubated at 37 °C overnight. The medium containing transfection reagents was replaced with fresh medium, and cells were incubated for an additional 24 h. The medium was collected, centrifuged (1,200 × g, 3 min), and filtered to retrieve the viral particles. The infection medium was prepared by adding polybrene (8 μg/mL) and the filtered supernatant containing the virus to a fresh, complete medium. The infection medium was added to a cell culture with 30-40% confluency, which had been seeded 24 h before the infection. The infection efficiency was evaluated using the inverted fluorescent microscope 72 h after infection. Pooled populations of cells expressing both α-tubulin-GFP and H2B-mCherry were established using the FACS Melody cell sorter (BD Biosciences). Ploidy of each newly established cell line was determined by a flow cytometry analysis of DNA content.

### Generation of stable cell lines using the random plasmid integration
The following procedure was used to generate stable U2OS cell lines overexpressing B55α-GFP. 40,000 U2OS cells were seeded in a 12-well plate. The following day, cells were transfected with a B55α-GFP expression vector using the Lipofectamine LTX reagent (Thermo-Fisher Scientific, A12621). A 100 μL transfection mixture (98 μL MEM, 10 μL lipofectamine LTX, 2 μL Plus reagent) containing 2 μg of plasmid DNA was prepared, incubated at room temperature for 5 min, and added dropwise to the cell culture. The following day, the medium containing transfection reagents was replaced with a fresh medium. Cells were grown for 7 days in a selection-free medium and an additional 7 days in a medium containing 1 mg/mL G418. Single GFP-positive cells were sorted using the FACS Melody cell sorter (BD Biosciences) and grown into clonal populations. Ploidy of each newly established cell line was determined by a flow cytometry analysis of DNA content.

### Generation of stable cell lines using the Sleeping Beauty transposon system
The following procedure was used to generate B55α-mAID-SMASh cells. An N-terminal SMASh-AID-B55 construct was cloned into a Sleeping Beauty transposon expression vector (Addgene, 60521) using Gibson assembly. The resulting plasmid, together with the Sleeping

Beauty transposase expression plasmid (Addgene, 34879), was transfected into RPE-1 OsTIR1 cells at a 10:1 ratio using Lipofectamine LTX (Thermo-Fisher Scientific, A12621). Transfected cells were grown for 7 days and then single-cell-sorted using blue fluorescent protein as a marker of successful integration. Single clones were verified by immunoblotting. The endogenous B55α was knocked out using CRISPR/Cas9 with gRNA targeting the PPP2R2A locus (5' CACCGGAG-GAGGGAATGATATTCAG 3'). The gRNA was cloned into the Cas9-T2A-GFP plasmid (Addgene, 48138) and transfected into B55α-mAID-SMASh-expressing RPE-1 cells using Lipofectamine LTX (Thermo-Fisher Scientific, A12621). Single cells were cloned by limiting dilution, and endogenous B55α knock-out was confirmed by immunoblotting and sequencing.

### Generation of stable RPE-1 p53-null cell line
The following procedure was used to generate RPE-1 p53-null cells. Cas9 ribonucleoprotein particles were prepared in the Cas9 buffer (20 mM HEPES pH 7.5, 150 mM NaCl, 2 mM MgCl$_2$, 1 mM TCEP) by mixing 120 pmol crRNA (5' UCGACGCUAGGAUCUGACUG 3'), 120 pmol tracrRNA (Merck, TRACRRNA05N) and 100 pmol His-Cas9-GFP. The mixture was incubated for 10 min at room temperature. RPE-1 cells were transfected using the Neon transfection system (Thermo Fisher) with a 10-μL tip (1,350 V per 20 ms per 2 pulses). Single cells were cloned by limiting dilution, expanded and validated by immunoblotting and PCR.

### Colony formation assay
60 mm dishes (Corning, 430166) were seeded with 240-720 cells in 3 mL of medium. Cells were allowed to attach for 5 to 6 h. Subsequently, 1 mL of fresh medium containing four times the desired concentration of Greatwall inhibitor C-604 was added to each dish. Cells were grown in the presence of C-604 for three days. The medium containing C-604 was removed and replaced with 4 mL of drug-free medium. Cells were grown for 7 to 14 days, depending on the cell line. Cells in each dish were fixed with 3 mL of 3.7% formaldehyde containing 1% methanol (Fisher Scientific, 10160052) in PBS for 15-20 min at 25 °C. Colonies were stained with 0.01% crystal violet (Sigma-Aldrich, V5265) in PBS for 30 min at 25 °C. The excess crystal violet was washed off with distilled H$_2$O. Dishes were air-dried. Stained colonies were imaged using the Epson Expression 1680 scanner. Colonies were counted semi-automatically using custom Cellpose[35] segmentation models. Four-parameter sigmoidal models and ED$_{50}$ estimates were generated using the drc package (version 3.0-1) in R.

### siRNA transfection
The siRNA transfections were performed using the Lipofectamine RNAi Max (Thermo Fisher Scientific, 13778100) and siRNAs smart pools (Dharmacon). For a single siRNA transfection in an immuno-fluorescence microscopy experiment, 12.5 μL of the siRNA transfection mixture (12.325 μL of MEM, 0.125 μL of RNAi Max, 0.05 μL of 20 μM siRNA) was prepared, gently mixed by pipetting and incubated at room temperature for 20-30 min. Subsequently, 37.5 μL of cell suspension containing 4,000 cells was mixed with 12.5 μL of siRNA transfection mixture and plated in a 96-well plate. After 24 h, the transfection medium was replaced with a fresh medium. For a single siRNA transfection in a colony formation experiment, 250 μL of the siRNA transfection mixture (246.5 μL of MEM, 2.5 μL of RNAi Max, 1 μL of 20 μM siRNA) was prepared, gently mixed by pipetting and incubated at room temperature for 20-30 min. Subsequently, 750 μL of cell suspension containing 80,000 cells was mixed with 250 μL of the siRNA transfection mixture and plated in a 24-well plate. After 24 h, the transfected cells were collected by trypsinisation, diluted to the desired density, and plated in a 60 mm dish, which was then processed as described in the colony formation assay protocol. A list of relevant siRNA sequences is included in Table 1.

**Table 1 | List of Smart pool siRNA sequences**

| Target | Smart pool ID | # | Sequence (5'-3') |
|---|---|---|---|
| Scrambled (CTR) Smart pool | D-001810-10-20 | 1 | UGGUUUACAUGUCGACUAA |
|  |  | 2 | UGGUUUACAUGUUGUGUGA |
|  |  | 3 | UGGUUUACAUGUUUUCUGA |
|  |  | 4 | UGGUUUACAUGUUUUCCUA |
| MASTL Smart pool | L-004020-00-0005 | 1 | GGACAAGUGUUAUCGCUUA |
|  |  | 2 | ACUGGACGCUCUUGUGUAA |
|  |  | 3 | GCAAAUUGUAUGCAGUAAA |
|  |  | 4 | CCAUUGAGACGAAAGGUUU |
| PPP2R2A Smart pool | L-004824-00-0005 | 1 | CAUACCAGGUGCAUGAAUA |
|  |  | 2 | GUAUAGAGAUCCUACUACA |
|  |  | 3 | GCAAGUGGCAAGCGAAAGA |
|  |  | 4 | AGACAUAACCCUAGAAGCA |
| PPP2R2B Smart pool | L-003022-00-0005 | 1 | UCGAUUACCUGAAGAGUUU |
|  |  | 2 | GGGUCGGGUUGUAAUAUUU |
|  |  | 3 | GAAUGCAGCUUACUUUCUU |
|  |  | 4 | CCACACGGGAGAAUUACUA |
| PPP2R2C Smart pool | L-019167-00-0005 | 1 | CGGAGGAUCUUUGCCAAUG |
|  |  | 2 | GAUACAACCUGAAGGAUGA |
|  |  | 3 | CCAACAACCUGUACAUCUU |
|  |  | 4 | GAAGAUUACCGAACGAGAU |
| PPP2R2D Smart pool | L-032298-00-0005 | 1 | GUAGGUCCUUCUUCUCAGA |
|  |  | 2 | UCGGAUAGCGCCAUCAUGA |
|  |  | 3 | GAGACUACCUGUCGGUGAA |
|  |  | 4 | GAGAACGACUGCAUCUUUG |
| NUAK1 Smart pool | E-004931-00 | 1 | CAACCAUCCUCAUAUCAUC |
|  |  | 2 | CCUUCAUGAUACUAGGUUU |
|  |  | 3 | UUAGCAUGUAGAUGGGAUA |
|  |  | 4 | UGAUUGAGUCUGUUUAGUA |
| MAP3K9 Smart pool | E-003585-00 | 1 | CGACCAUCUUUCACGAAUA |
|  |  | 2 | UCCUUAACUUGAAUGUUGA |
|  |  | 3 | UGGUUAUGUUUGGUUAGUA |
|  |  | 4 | CCAGCAAGUAUUGAGGGUA |
| MELK Smart pool | E-004029-00 | 1 | CUAUCUAGCUGCAAGGUAU |
|  |  | 2 | CCAAAGACUCCAGUUAAUA |
|  |  | 3 | GACUAAAGCUUCACUAUAA |
|  |  | 4 | GUAUGAAUCUAAAUCAAGC |
| ULK2 Smart pool | E-005396-00 | 1 | CUAUGAUGUUCAGGAAUUA |
|  |  | 2 | CCUCAAGACUUAAGGAUGU |
|  |  | 3 | GUGUAAAGGUAAAUGUAUU |
|  |  | 4 | GCAUUAUUCUUGAAACUAG |
| HIPK2 Smart pool | E-003266-00 | 1 | GUGGGAAAUCUAUGGUUUU |
|  |  | 2 | CCAGUACCCUUACAUAUAA |
|  |  | 3 | GCGACAUGUUGGUAGAAAA |
|  |  | 4 | GUGUUAUUGCAGAAUUGUU |
| CHEK1 Smart pool | E-003255-00 | 1 | GUAGUAAAAUUCUAUGGUC |
|  |  | 2 | GGUUUAUCUGCAUGGUAUU |
|  |  | 3 | GGGAUAACCUCAAAAUCUC |
|  |  | 4 | GGAUGAUAAAAUAUUGGUU |

### RNA extraction and reverse transcription

Total RNA was extracted using the total RNA miniprep kit (Monarch, T2010S). RNA yields were estimated using NanoDrop 1000 (Thermo Fisher). Reverse transcription was performed using the First-Strand cDNA Synthesis Kit (Cytiva, GE27-9261-01) according to the manufacturer's instructions. The following protocol describes the

processing of 1 sample. Purified RNA (200 ng) was diluted with RNase-free $H_2O$ to a final volume of 20 μL. This mixture was incubated at 65 °C for 10 min and then placed on ice. The RNA was mixed with 11 μL of cDNA reaction mix, 1 μL of 200 mM DTT and 1 μL of random hexamer primers (0.2 μg/μL). No reverse transcriptase (No-RT) control reaction omitted the cDNA reaction mix. Reverse transcription was carried out at 37 °C for 60 min and subsequently terminated at 72 °C for 5 min. cDNA was stored at -80 °C.

### Quantitative PCR

Quantitative PCR (qPCR) was performed using the Luna Universal qPCR Master Mix (NEB, M3003E) and the AriaMx Real-Time PCR System (Agilent Technologies). The following protocol describes the processing of 1 sample. A 20 μL qPCR reaction was prepared by mixing 10 μL of Luna universal qPCR mix, 0.5 μL of forward primers (10 μM), 0.5 μL of reverse primers (10 μM), 1 μL of cDNA and 8 μL of nuclease-free $H_2O$. No template control (NTC) reactions did not contain the cDNA template. No-RT reactions were templated by the control no-RT sample. For every unique qPCR reaction, three technical replicates were prepared, analysed and averaged. A list of relevant qPCR primers is included in Table 2.

### Immunofluorescence

Cells were fixed with 4% formaldehyde (CST, 47746) for 15 min at 25 °C and washed once with 100 μL of PBS. Cells were permeabilised and blocked with the blocking solution (CST, 12411) for 60 min at 25 °C and washed once with 100 μL of PBS. If the experiment involved EdU labelling, cells were stained with the Click-iT EdU cell proliferation kit for imaging (Invitrogen, C10340) according to the manufacturer's instructions. Cells were washed three times with 100 μL of PBS and incubated overnight at 4 °C with the antibody dilution buffer (CST, 12378) containing the primary antibodies. Cells were washed three times with 100 μL of PBS and incubated with the antibody dilution buffer containing secondary antibodies for 30 min at 25 °C. Cells were washed three times with 100 μL of PBS and incubated in PBS containing 1 μg/mL of 4',6-diamidino-2-phenylindole (DAPI) for 30 min at 25 °C. Cells were washed once with 100 μL of PBS and covered with 200 μL of PBS. Stained samples were stored at 4 °C.

### Microscopy

For immunofluorescence and live cell imaging applications, cells were grown in black 96-well plates with an optically clear flat bottom (Revvity, 6055302). Imaging was done using the Operetta CLS high-content analysis system (PerkinElmer). Live cell imaging was performed at a constant temperature of 37 °C and 5% $CO_2$.

### Protein extraction and Western blotting

Cells were collected by trypsinisation or mitotic shake-off and washed once with 1–2 mL of PBS. Depending on the size of the cell pellet, cells were resuspended in 20–100 μL of 9 M urea supplemented with 5% 2-mercaptoethanol. Cell lysates were sonicated and incubated at 95 °C for 5 min. Protein concentrations were determined with the Bradford reagent (MERCK, B6916). Protein samples were stored at -20 °C. Protein samples were mixed with 4× Laemmli sample buffer (Bio-Rad, 1610747) and incubated at 95 °C for 2 min. Depending on the application, 5-20 μg of total protein were resolved by a 10%, 12% or 13% SDS-polyacrylamide gel. Proteins were transferred onto a PVDF membrane (Amersham, 10600023) using the Trans-Blot Turbo Transfer System (Bio-Rad) at a constant 25 V for 30 min. The transfer sandwich was assembled as follows: three sheets of filter paper soaked in the anode one buffer (300 mM Tris, 20% methanol, pH 10.4), two sheets of filter paper soaked in the anode two buffer (25 mM Tris, 20% methanol, pH 10.4), membrane soaked in methanol, gel with size-resolved proteins, and five sheets of filter paper soaked in cathode buffer (25 mM Tris, 40 mM 6-aminohexanoic acid, 20% methanol, pH 9.4). The membrane

**Table 2 | List of qPCR primers**

| Target | Forward sequence (5'-3') | Reverse sequence (5'-3') |
|---|---|---|
| GAPDH | CCATCTTCCAGGAGCGAG | GACTCCACGACGTACTCAG |
| ACTB | GACCCAGATCATGTTTGAGAC | GATAGCACAGCCTGGATAG |
| MASTL | GCTCGTTGGGATTTAACAC | GACATAGGGCATACGAGTC |
| PPP2R2A | AGACAGGAGTTTTAACATTGTG | GTGTTACAGCTGTTTGGATG |
| PPP2R2B | GAGGACATTGATACCCGCAA | ACCCTGATTTTTACTCTCCTGCT |
| PPP2R2C (#1) | TGACAAGCATTCCAAGCTC | GACACGGAGGAGATGATTTC |
| PPP2R2C (#2) | GCTATGTGACTGAAGCTGAC | TTCCCGCTGGAAGATGAC |
| PPP2R2D | CTGTCCACCAACGATAAAAC | CTCTTCATCCTTCAGGTTGT |
| NUAK1 | CGAGTGGTTGCTATAAAATCC | ATCTCTCGTCTGATGTGAAC |
| MAP3K9 | CCGCCCATTCAGTTGTTAG | ATCCAGAAAGCACGATAGAC |
| MELK | ATATCTTGGATCAGAGGCAG | GGTAGAAATCCACACATAAGAAC |
| HIPK2 | CCTTTGTGGAGACTGAAGAC | TGGGCCATATCATCTAAACAG |
| CHEK1 | CTCAAGAAAGGGGCAAAAAG | ATGTGCTTAGAAAATCCACTG |
| ULK2 | ATGGCTCCTGAGGTTATTATG | CTGTTCCTATGCTCCACAAG |

with transferred proteins was rinsed with methanol and washed with distilled H$_2$O. The membrane was stained with 0.1% (w/v) Ponceau S (Sigma-Aldrich, P3504) in 5% acetic acid and imaged. The membrane was destained by distilled H$_2$O and blocked in 3% BSA in TBST (20 mM Tris, 150 mM NaCl, 0.1% Tween 20, pH 7.6) for 60 min at 4 °C. The membrane was subsequently incubated in 3% BSA in TBST containing primary antibodies overnight at 4 °C. The membrane was washed three times with TBST (5 min, 5 min, 15 min) and incubated in 3% BSA in TBST containing HRP-fused secondary antibodies for 60 min at 4 °C. The membrane was washed three times with TBST (5 min, 5 min, 15 min), then covered with the Clarity Western ECL substrate (Bio-Rad, 1705061), and imaged using the ImageQuant LAS 4000 (GE Healthcare) or the ImageQuant 800 (Amersham) system. For the detection of ENSA/ARPP19 and phospho-ENSA(S67)/ARPP19(S62), the ECL substrate was supplemented with 10% SuperSignal West Femto Maximum Sensitivity Substrate (Thermo Scientific, 34095). Protein intensities were quantified using the ImageJ software[60]. Dose-dependent changes in protein intensities were analysed using the drc package (version 3.0-1) in R (version 4.2.3).

**Antibodies**

For Western blots, the following primary antibodies were used: anti-GWL (1:1,000; Sigma-Aldrich, HPA027175), anti-PPP2R2A (1:1,000; Santa Cruz, sc-81606), anti-ENSA/ARPP19 (1:1,000; Abcam, ab180513), anti-ENSA(pS67)/ARPP19(pS62) (1:1,000; CST, 5240), anti-PPP2C (1:1,000; CST, 2038), anti-β-actin (1:4,000; Abcam, ab3280), anti-GAPDH (1:20,000; Abcam, ab8245). For Western blots, the following secondary antibodies were used: anti-mouse HRP-conjugated, raised in goat (1:5,000; Dako, P0447), and anti-rabbit HRP-conjugated, raised in goat (1:2,000; Dako, P0448). For immunofluorescence, the following primary antibodies were used: anti-α-tubulin (1:1,000; CST, 3873) and anti-p21 (1:800; CST, 2947). For immunofluorescence, the following secondary antibodies were used: anti-mouse Alexa Fluor 488 conjugate (1:2,000, CST, 4408S) and anti-rabbit Alexa Fluor 555 conjugate (1:2,000, CST, 4413S).

**Sample preparation for phosphoproteomics analysis**

Cell pellets were lysed with a Urea buffer (8 M urea in 20 mM in HEPES, pH 8.0 supplemented with 1 mM Na$_3$VO$_4$, 1 mM NaF, 1 mM Na$_2$H$_2$P$_2$O$_7$ and 1 mM sodium β-glycerophosphate), sonicated for 30 cycles (30 s on 30 s off) in a Diagenode Bioruptor® Plus and insoluble material was removed by centrifugation (13,000 × g for 10 min at 4 °C). Protein concentration was determined using the BCA Protein Assay Kit. 110 μg of extracted proteins in a final volume of 200 μL were reduced with dithiothreitol (DTT, 10 mM) for 1 h at 25 °C, and alkylated with Iodoacetamide (IAM, 16.6 mM) for 30 min at 25 °C. Then, samples were diluted with 20 mM HEPES (pH 8.0) to a final concentration of 2 M urea and digested with equilibrated trypsin beads (50% slurry of TLCK-trypsin) overnight at 37 °C. The equilibration of the beads was performed by washes with 20 mM HEPES (pH 8.0). After digestion and trypsin bead removal by centrifugation (2000 × g for 5 min at 5 °C), peptide solutions were transferred into 96-well plates and acidified by adding trifluoroacetic acid (TFA) to a final concentration of 0.1%. Then, samples were desalted and subjected to phosphoenrichment using the AssayMAP Bravo (Agilent Technologies) platform. For desalting, the protocol peptide clean-up v3.0 was used. Reverse phase S cartridges (Agilent, 5 μL bed volume) were primed with 250 μL 99.9% acetonitrile (ACN) with 0.1% TFA and equilibrated with 250 μL of 0.1% TFA at a flow rate of 10 μL/min. The samples were loaded (770 μL) at a rate of 20 μL/min, followed by an internal cartridge wash with 250 μL of 0.1% TFA at a flow rate of 10 μL/min. Peptides were then eluted with 55 μL of 70% ACN, 0.1% TFA, into 96-well protein LoBind plates containing 50 μL of 1 M glycolic acid with 50% ACN and 5% TFA. Following the Phospho Enrichment v 2.1 protocol, phosphopeptides were enriched using 5 μL Assay MAP TiO2 cartridges on the Assay MAP Bravo platform. The cartridges were primed with 100 μL of a 5% ammonia solution in 15% ACN at a flow rate of 300 μL/min and equilibrated with 50 μL of loading buffer (1 M glycolic acid in 80% ACN, 5% TFA) at 10 μL/min. Samples eluted from the desalting were loaded onto the cartridge at a flow rate of 3 μL/min. The cartridges were washed with 50 μL of loading buffer, and phosphopeptides were eluted with 25 μL of a 5% ammonia solution containing 15% ACN directly into 25 μL of 10% formic acid. Phosphopeptides were lyophilised in a vacuum concentrator and stored at -80 °C. This protocol is analogous to the previously reported procedure[61,62].

**LC-MS/MS analysis**

Phosphopeptides were re-suspended in 8 μL of reconstitution buffer and sonicated for 2 minutes at RT. After centrifugation (13,000 × g, 5 min, 4 °C), 5 μL was loaded onto an LC-MS/MS system. The system consisted of a nano-flow ultra-high-pressure liquid chromatography system, UltiMate 3000 RSLC nano (Dionex), coupled to a Q Exactive Plus using an EASY-Spray system. The LC system used mobile phases A (3% ACN in 0.1% formic acid) and B (100% ACN in 0.1% formic acid). Peptides were loaded onto a μ-pre-column and separated in an analytical column. The gradient: 1% B for 5 min, 1% B to 35% B over 90 min. Following elution, the column was washed with 85% B for 7 min and then equilibrated with 3% B for 7 min at a flow rate of 0.25 μL/min. Peptides were nebulised into the online-connected Q-Exactive Plus system, operating with a 2.1 s duty cycle. Acquisition of full scan survey

spectra (m/z 375–1500) with a 70,000 FWHM resolution was followed by data-dependent acquisition in which the 15 most intense ions were selected for HCD (higher energy collisional dissociation) and MS/MS scanning (200–2000 m/z) with a resolution of 17,500 FWHM. A 30 s dynamic exclusion period was enabled with an exclusion list with a 10 ppm mass window. Overall duty cycle generated chromatographic peaks of approximately 30 s at the base, allowing the construction of extracted ion chromatograms (XICs) with at least ten data points.

## Phosphopeptides identification and quantification

Peptide identification from MS data was automated using a Mascot Daemon (v2.8.0.1) workflow in which Mascot Distiller generated peak list files (MGF) from RAW data, and the Mascot search engine matched the MS/MS data stored in the MGF files to peptides using the SwissProt Database restricted to Homo sapiens (SwissProt_2021_02.fasta). Searches had an FDR of ~1% and allowed 2 trypsin missed cleavages, mass tolerance of ± 10 ppm for the MS scans and ± 25 mmu for the MS/MS scans, carbamidomethyl Cys as a fixed modification and oxidation of Met, PyroGlu on N-terminal Gln and phosphorylation on Ser, Thr, and Tyr as variable modifications. Pescal was used for label-free quantification of the identified peptides[62,63]. The software constructed XICs for all the peptides identified in at least one of the LC-MS/MS runs across all samples. XIC mass and retention time windows were ± 7 ppm and ± 2 min, respectively. Quantification of peptides was achieved by measuring the area under the peak of the XICs. Individual peptide intensity values in each sample were normalised to the sum of the intensity values of all the peptides quantified in that sample. Finally, the phosphoproteomics data were processed and analysed using a public bioinformatic pipeline developed in an R environment (https://github.com/CutillasLab/protools2/). The normalised data was centred, $\log_2$ scaled, and 0 values were introduced using the minimum feature value in the sample minus one. Then, statistical differences (p-values) were calculated using LIMMA[64], and then adjusted for FDR using the Benjamini-Hochberg procedure. Differences were considered statistically significant when p-values < 0.05 and FDR < 0.1. Analysis of differentially phosphorylated peptides was performed in R (version 4.2.3) using custom scripts (https://github.com/R-Zach/Cellular-responses-to-Greatwall-inhibition.git).

## Synthesis of MKI-2

MKI-2 was synthesised following a two-step procedure from commercially available 2,4-dichloroquinazoline. Substitution using 5-amino-3-cyclopropyl-1H-pyrazole led to the formation of 2-chloro-N-(5-cyclopropyl-1H-pyrazol-3-yl)quinazolin-4-amine, which was further reacted with 4-aminophenylacetonitrile to give MKI-2. To produce 2-chloro-N-(5-cyclopropyl-1H-pyrazol-3-yl)quinazolin-4-amine, a suspension of 2,4-dichloroquinazoline (250 mg, 1.26 mmol), 3-amino-5-cyclopropyl-lH-pyrazole (155 mg, 1.26 mmol) and triethylamine (0.21 mL, 1.51 mmol) in tetrahydrofuran (7.5 mL) was stirred at room temperature overnight (18 h). The solid was collected by filtration, washed with EtOH (approximately 5 mL), then dried under vacuum at 40 °C overnight (18 h) to afford 2-chloro-N-(5-cyclopropyl-1H-pyrazol-3-yl)quinazolin-4-amine (345 mg, 0.72 mmol, 58% yield) as a white solid. The product was characterised by $^1$H NMR (600 MHz, DMSO-$d_6$) with signals at δ 12.36 (s, 1H), 10.80 (s, 1H), 8.64 (d, J = 8.4 Hz, 1H), 7.85 (t, J = 7.6 Hz, 1H), 7.69 (d, J = 8.3 Hz, 1H), 7.57 (t, J = 7.7 Hz, 1H), 6.50 (d, J = 2.0 Hz, 1H), 2.02 – 1.92 (m, 1H), 0.99 – 0.93 (m, 2H), 0.77 – 0.70 (m, 2H). To synthesise MKI-2, concentrated hydrochloric acid (0.35 mL, 1.57 mmol) was added to a suspension of 2-chloro-N-(5-cyclopropyl-1H-pyrazol-3-yl)quinazolin-4-amine (75 mg, 0.16 mmol) and 4-Aminophenylacetonitrile (31 mg, 0.24 mmol) in t-butanol (3.75 mL). The resulting reaction mixture was stirred at 90 °C for 72 h. At room temperature, MeOH (5 mL) was added, then the solid was collected by filtration and rinsed twice with MeOH (2 × 1 mL). The solid was collected and dried under vacuum at

40 °C overnight (18 h) to afford [4-({4-[(5-cyclopropyl-1H-pyrazol-3-yl)amino]quinazolin-2-yl}amino)phenyl]acetonitrile MKI-2 (24 mg, 0.06 mmol, 38% yield) as an off-white solid. MKI-2 was characterised by $^1$H NMR (600 MHz, DMSO-$d_6$) with signals at δ 13.08 (s, 1H), 11.57 (s, 1H), 10.61 (s, 1H), 8.67 (d, J = 8.3 Hz, 1H), 7.87 (t, J = 7.8 Hz, 1H), 7.64 (d, J = 8.3 Hz, 1H), 7.55 (d, J = 7.9 Hz, 2H), 7.50 (t, J = 7.9 Hz, 1H), 7.44 (d, J = 8.0 Hz, 2H), 6.35 – 6.02 (br s, 1H), 4.10 (s, 2H), 1.86 (s, 1H), 0.99 – 0.93 (m, 2H), 0.63 – 0.50 (m, 2H). The LC-MS analysis (14 min, acidic method) showed a retention time of 3.77 min with m/z = 382 [(M + H)$^+$, ESI$^+$] and m/z = 380 [(M−H)$^-$, ESI$^-$], and a purity of 99% (UV at 254 nm).

## Image processing and analysis

Raw immunofluorescence imaging data were stored on the OMERO cloud repository[65]. Cells and nuclei were segmented using Cellpose[35] (version 2.0.5) and custom-built segmentation models as part of a custom-made automated image analysis pipeline (https://github.com/HocheggerLab/Omero_Screen.git). Integrated DAPI and mean nuclear EdU and p21 intensities were normalised using values representing the first peaks of respective intensity distributions in untreated cell populations. Downstream analysis and quantifications were done using Python (version 3.9.13) or R (version 4.2.3) and custom-built scripts (https://github.com/R-Zach/Cellular-responses-to-Greatwall-inhibition.git).

## Mathematical modelling

Cellular sensitivity to GWL inhibition was modelled based on expression levels of GWL, B55α, PPP2Cα/β and ENSA/ARPP19, using linear modelling without weighting. Optimal parameters for gradient and intercept were estimated using ordinary least squares regression. Linear regression and model fitting were performed with the fitdistrplus package (version 1.2-1) in R (version 4.4.0).

## Structural modelling

Compound **21** (C-604) was docked into an AlphaFold model of human GWL [AF-Q96GX5-F1] using Flare (academic version, courtesy of Cresset Inc., UK). The top-ranking pose was subsequently energy-minimised using the inbuilt functionality of the software (AMBER-GAFF2 forcefield)[66]. Molecular graphic figures were prepared using PyMOL (v3.10).

## Public data analysis

Cancer cell line RNA expression and gene dependency data[51] were retrieved from the DepMap repository (https://depmap.org/portal). The cancer tissue RNA expression analysis was based on data generated by the TCGA research network (https://www.cancer.gov/tcga). Public data were analysed in R (version 4.2.3) using custom-built scripts (https://github.com/R-Zach/Cellular-responses-to-Greatwall-inhibition.git).

## Statistical analysis

Statistical significance was determined by an unpaired two-tailed t-test (normally distributed data) or Wilcoxon signed-rank test (not normally distributed data). Normality of the tested samples was determined by the Shapiro-Wilk test. When appropriate, p-values were corrected for multiple testing using the Benjamini-Hochberg procedure. The exact p-values are included in the Source data file. Statistical analysis was performed in R (version 4.2.3).

## Biological materials

This study reports the production and evaluation of a small-molecule tool compound C-604 (UOS-00054604). Details of the synthesis and activity of the compound are provided in the manuscript. Compound MKI-2 was synthesised according to the published protocols[31]. All compounds were validated by $^1$H NMR and LC-MS.

## Inclusion and ethics

This study was conducted in accordance with the ethics and inclusion guidelines provided by the University of Sussex: https://www.sussex.ac.uk/about/culture-equality-and-inclusion and https://www.sussex.ac.uk/staff/research/governance.

## Reporting summary

Further information on research design is available in the Nature Portfolio Reporting Summary linked to this article.

## Data availability

Raw mass spectrometry proteomics data have been deposited in the ProteomeXchange Consortium via the PRIDE[67] partner repository with the accession code PXD059047 and can be retrieved from https://www.ebi.ac.uk/pride/archive/projects/PXD059047. Additional processed datasets can be retrieved from: https://doi.org/10.6084/m9.figshare.28829927 (phosphoproteomics), https://doi.org/10.6084/m9.figshare.28829816 (immunofluorescence microscopy), https://doi.org/10.6084/m9.figshare.28848074 (The Cancer Genome Atlas RNA expression data), https://doi.org/10.6084/m9.figshare.28829915 (live-cell imaging), https://doi.org/10.6084/m9.figshare.28829903 (quantification of colony formation) and https://doi.org/10.6084/m9.figshare.28829306 (quantification of Western blots). Source data are provided with this paper.

## Code availability

Custom code can be obtained from https://github.com/HocheggerLab/Omero_Screen.git and. https://github.com/R-Zach/Cellular-responses-to-Greatwall-inhibition.git [68].

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

## Acknowledgements

This work was supported by funding from CRUK - C28206/A14499 (R.Z., S.M.G., M.M., S.D., P.R.C., H.H.), the Wellcome Trust 110578/Z/15/Z (M.A., K.O., N.P., W.P., J.B., M.R., C.T., T.R., R.W., H.P., D.L.G., A.W.O., S.E.W.); 205150/Z/16/Z (K.H.C., H.Y.); 225852/Z/22/Z (J.H.) and the Ministry of Higher Education, Saudi Arabia (A.A.). Characterisation of CHOL cell line models was supported by the Mahidol Sussex Seed Fund (K.V., D.D.).

## Author contributions

R.Z. designed and executed experiments and wrote the manuscript; M.A., K.O., N.P., W.P., J.B., M.R., C.T., T.R., R.W., D.C., S.O., J.S., H.P., D.L.G., S.E.W. carried out small molecule synthesis and SAR studies; S.M.M.G., P.R.C. performed mass spectrometry analysis; M.M., A.A., L.C.E., S.D. and W.R.F. performed cell biological and biochemical experiments; A.D.H. performed image-analysis; K.V., D.D. developed CHOL cell line models; O.B. generated B55α degron RPE-1 cells; K.B. generated RPE-1 p53-null cells; K.H.C. and H.Y. performed experiments in *Xenopus* egg extracts; J.A.H. performed mathematical modelling; D.M.A. and T.A.H. performed small molecule screen; A.W.O. performed protein purification, assay establishment and structural modelling; H.H. designed experiments and wrote the MS.

## Competing interests

D.M.A. and T.A.H. are employees of AstraZeneca. D.M.A. and T.A.H. are AstraZeneca shareholders. All other authors declare no competing interests.

## Additional information

Róbert Zach [1], Michael Annis[2,9], Sandra M. Martin-Guerrero [3,9], Abdulrahman Alatawi [1,9], Kim Hou Chia [4], Megan Meredith[1], Kay Osborn[2], Nisha Peter[2], William Pearce[2], Jessica Booth[2], Mohan Rajasekaran[2], Samantha Dias[1], Lily Coleman-Evans[1], William R. Foster[1], Jon A. Harper[1], Alex D. Herbert [1], Catherine Tighe[2], Tristan Reuillon[2], Ryan West [2], Oliver Busby[1], Kamila Burdova[1], Damien Crepin[2], Sergi Ortoll[2], Kulthida Vaeteewoottacharn[5], Donniphat Dejsuphong[6], John Spencer [2], Hitesh Patel[7], Darren Le Grand[7], Thomas A. Hunt [8], David M. Andrews [8], Hiroyuki Yamano[4], Pedro R. Cutillas [3], Antony W. Oliver [1], Simon E. Ward [7] & Helfrid Hochegger [1]

[1]Genome Damage and Stability Centre, School of Life Sciences, University of Sussex, Brighton BN1 9RQ, UK. [2]Sussex Drug Discovery Centre, School of Life Sciences, University of Sussex, Brighton BN1 9QJ, UK. [3]Barts Cancer Institute, Queen Mary University of London, London EC1M 6BQ, UK. [4]University College London Cancer Institute, London WC1E 6DD, UK. [5]Department of Biochemistry, Faculty of Medicine, Khon Kaen University, Khon Kaen 40002, Thailand. [6]Program in Translational Medicine, Chakri Naruebodindra Medical Institute, Faculty of Medicine Ramathibodi Hospital, Mahidol University, Samutprakarn 10540, Thailand. [7]Medicines Discovery Institute, Cardiff University, Cardiff CF10 3AT, UK. [8]Oncology Chemistry, Oncology Targeted Discovery, AstraZeneca, Cambridge CB2 0AA, UK. [9]These authors contributed equally: Michael Annis, Sandra M. Martin-Guerrero, Abdulrahman Alatawi. ✉e-mail: r.zach@sussex.ac.uk; h.hochegger@sussex.ac.uk

