## [Transparent Peer Review file · Nature Communications]

The Balance between B55 α and Greatwall expression levels predicts sensitivity to Greatwall inhibition in cancer cells

Corresponding Author: Dr Robert Zach

A version of this paper was originally rejected for publication by Nature Communications, however that decision was reconsidered after appeal by the authors.

Version 0:

Reviewer comments:

Reviewer #1

(Remarks to the Author)

This work develops a small molecule inhibitor (C-604) of the MASTL/Greatwall kinase and provides extensive characterization of its cellular effects. MASTL is an inhibitor of the PP2A-B55 phosphoprotein phosphatase, through the phosphorylation of ENSA/ARPP19, and a major regulator of mitotic entry. MASTL is an oncogene and a possible relevant target of cancer therapies. To this end the authors here develop a small molecule inhibitor of MASTL and provide extensive in vitro and in vivo characterization. Overall, the inhibitor shows high level of specificity and induces the expected cellular phenotypes, mitotic entry and mitotic phenotypes, but the severity of these phenotypes appears to depend on the cell lines analysed which might partly be due to p53 status. Interestingly the cellular toxicity of C-604 is dependent on B55 α status, the major isoform of B55 subunits, and thus cellular toxicity is due to overactivation of PP2A-B55. The authors perform extensive cellular characterization, phosphoproteomic analysis and outlines possible cancers that could be particularly relevant for targeting with C-604.

Overall this is a really nice study that I recommend for publication in Nature Communications and only have minor suggestions for improvement.

- 1) I think the data in Figure S1 and S2 is really important, and I suggest to move this to a main figure 1.
- 2) The model for how C-604 engages MASTL (Fig. S1C) have the authors analyzed this experimentally? If so, would be interesting to include.
- 3) The authors observe inhibition of other kinases in the in vitro screens but claim that cellular phenotypes are not due to these kinases – it was a little unclear to me how they reached this conclusion. Could the please explain/expand.
- 4) In the phosphoproteomic analysis they compare their data with that of data from the Barr lab generated by a different approach and see little overlap. Have they compared to other PP2A-B55 data sets (Hein et al 2023, Kruse et al 2020, Swartz 2023)?

(Remarks on code availability)

Reviewer #2

(Remarks to the Author)

The study by Zach et al described 1) the development of GWL inhibitor; 2) its impact on mitotic events and 3) expression level of B55 α being a predictor of cellular sensitivity to GWL inhibitor. The first two points have been established in several previous literatures. For example, a number of study by Dr. Jae-Sung Kim's group have produced enhanced GWL inhibitors that appear much more potent than the compound reported here, both in enzymatic assays and colony formation assays (PMID 34358073). Several other groups also published GWL/MASTL inhibitors. The observation that MASTL inhibition lead to mitotic catastrophe has also been established in several studies (e.g. PMID 29976159). The last point, that expression level of B55 α determines MASTL inhibitor sensitivity seems to be also covered by PMID 29976159-- "we found that MASTL-depletion-induced mitotic cell death was partially blocked by PP2A inhibition from PP2A-A α / β siRNA (Fig. 5i)." Considering

existing literatures, the advance of knowledge gained from this study is not very significant.

Specific points:

1. The major argument is that different B55a levels in cancer cells determines sensitivity to GWL inhibitors (Figure 6E). However it appears that the overall trend received significant contribution from 2 cell lines (BT549, the most sensitive and HCC1143 the most resistant). If these two cell lines are taken out, it would be rather difficult to establish a trend using data from remaining cell lines. High reliance on two data points on the most extreme end is a concern.
2. As a drug discovery paper, IC50/EC50 numbers should be clearly stated in main text.
3. Previous studies on similar conclusions e.g. PMID 29976159 should be cited.

(Remarks on code availability)

Reviewer #3

(Remarks to the Author)

In their present study, Zach et al. introduce C-604, a novel Greatwall (GWL) inhibitor and analyze its biological effects in several cancerous and noncancerous cell lines. Their findings reveal that GWL inhibition leads to destabilization of the mitotic phosphoproteome, resulting in premature mitotic exit and defective cytokinesis. Furthermore, they confirm that the defects caused by low GWL activity are due to its inhibitory interactions with the phosphatase PP2A-B55 α and demonstrate that sensitivity to GWL inhibition is determined by the ratio of GWL and PP2A- B55 α levels in a given cell line.

The paper is overall well written and the clear visualization of results allows easy access to the data. The authors present a comprehensive analysis of the cellular effects of pharmacological GWL inhibition, employing different methods and testing multiple mammalian cell lines, which supports the generalizability of their findings. Experiments are mostly well controlled and sufficiently replicated. While several small-molecule GWL inhibitors have been published in recent years as mentioned in the manuscript, the authors argue that previous studies lack a sufficient characterization of their biological mode of action. With the introduced selective and highly affine GWL inhibitor C-604, they successfully reproduce the full spectrum of known phenotypic consequences of loss of GWL activity and provide phosphoproteomic data to gain mechanistic insights. Importantly, the authors demonstrate specificity of their compound, as the observed effects were reversed upon knockdown of the GWL substrate PP2A-B55 α . Overall, the study provides a valuable new tool compound and a comprehensive data set to the field, although new insights are limited due to the confirmatory nature of most results. Furthermore, the analyses indicates that GWL inhibition is of limited therapeutic relevance.

Before publication, the authors need to address the following points to strengthen the manuscript and correct some shortcomings:

Major points:

1. It is laudable that the authors aim to comprehensively assess off-target effects of the C-604. However, these issues should be further addressed:
 - a. There seems to be limited correlation between the in vitro measured activity of the kinase panel (Fig. S1d) and the cellular IC50 measurements of the selected candidates (Fig. S1e). The selection of kinases with >80% inhibition for further testing therefore seems arbitrary. There might well be additional kinases with cellular IC50 < 50nM, which would need to be included in off target testing. The authors should expand the measurement of cellular IC50 values to identify such candidates.
 - b. This is specifically important as there are kinases with known mitotic phenotypes among the top 20 hits from the in vitro screening, such as the checkpoint kinase CHEK2 (PMID: 15048074, PMID: 35851582, PMID: 38559033). Kinases with known mitotic phenotypes should be tested for cellular IC50 values and if applicable included in further testing.
 - c. Testing for off target effects using phenotypic reversal upon siRNA is overall convincing. However, the authors performed these tests only at 0.5 μ M C-604 concentration (Fig. S7c-d), while most of the phenotypic studies for GWL inhibition were done at up to 2 μ M (Fig. 1-4). As the observed effects were most obvious at high concentrations, it would be advisable to repeat the off-target kinase testing at 2 μ M C-604 as well. Also, some knockdowns show strong cell cycle alterations already in control condition (e.g. NUAK, MELK and CHEK1 siRNA abolishing S-phase populations in RPE-1 cells). The author should comment to which extent these alterations affect conclusions about phenotypic outcomes of GWL inhibition. If cells are not dividing, mitotic defects will not be assessable.
2. The authors suggest that the phenotypic outcome of GWL inhibition is only partially dependent on p53 (page 8, line 205-214). This finding is interesting and relevant. However, its validity is hard to assess as the data in Figure S5 is missing. Instead, the figure panels shown are a replication of Fig. 5. Moreover, the authors should further discuss which p53-independent pathways could be involved in the physiological response to GWL inhibition.
3. The statistical analysis of the data should be reviewed as the authors apply t-test for all data sets. The applicability of t-test should be confirmed by assessing normality of the underlying distributions, and corrections for multiple testing should be applied where necessary. It might be helpful to include analysis of effect sizes, e.g. by calculating Cohen's d values.
4. To allow evaluation of the dependency for PP2A-B55 α , the extent and specificity of the knockdown of all subunits should be shown, optimally at the protein level (Fig. 3).
5. The model for predicting sensitivity to GWL inhibition should be experimentally tested with cell lines not used for model generation (Fig. 6).
6. Please ensure that all experimental procedures are fully described in the methods section. For example, currently only sequences of siRNAs are given, but no information about transfection procedures and concentrations used. Also, please expand legends to include all important information (assays and conditions used) and explain normalization and visual

display of the data. Some legends miss descriptions of entire figure panels (e.g. Fig. 8d-f)!

Minor points:

- The initial screen of the AstraZeneca compound collection is described in the first paragraph of the results section, but it is hard to assess without any representation of the data. For example, six clusters are mentioned from which compound 1 emerged as a candidate without giving any information about them.
- Could the authors explain why two Conditional Degron Tags (mAID and SMASh) were used to degrade PP2A-B55 α ?
- Please add information of how many cells were analyzed for all data sets (missing e.g. for Figure 2d, Figure 3, Figure S6).
- Please make sure that abbreviations are spelled out at first time of usage, which might be as part of a supplemental data figure, to help readers not familiar with these kind of measurements / analysis.
- Figure 3b: It might be helpful to not only statistically test changes of cell cycle distributions to the 0 μ M condition for each siRNA, but also compare the corresponding conditions to the siCTR condition.
- Figure 4d: Euclidean distance was calculated in Figure 2g, so the used Euclidean values correspond to Figure 2g, not Figure 3f.
- Figure 4g/h: As values seem discrete, could it be that median, not means are represented by red lines?
- Figure 4f: Please add units for the concentration of C-604.
- Figure 5 e and f: Legend should explain difference between Figure 5e and Figure 5f.
- Figure S1b: It would be nice to show the experimental data used to estimate IC50 values as in Fig. S1e.
- Figure S2a: The choice of concentrations seems suboptimal for generating a valid fit. Further information about how the in vitro assay was performed and how the phosphorylation of ENSA was finally measured would be helpful for the reader.
- Lines 117-124: The reference to Figures S1 d,e,f is missing.
- Line 216: The reference to Figure S2d is wrong, it should be Figure S2e.
- Lines 228-229: The reference to Figure S2d is wrong, it should be Figure S2e.
- Line 240: The reference to Figure 5S is wrong, it should be Figure S6
- Please add page numbers in a revised manuscript.

(Remarks on code availability)

The code was only briefly reviewed and seemed reasonable. On first glance, I didn't find the respective data files, so I could not reproduce the analysis.

Reviewer #4

(Remarks to the Author)

(Remarks on code availability)

Version 1:

Reviewer comments:

Reviewer #1

(Remarks to the Author)

The authors have done a good job in addressing my concerns and that of the other reviewers. In particular the increased characterisation showing on-target has improved the paper. They have also done a good job in addressing the novelty concern raised by reviewer 2 by showing that previous inhibitors likely have off-target effects. I recommend publication. Small point is that Krus et al should be Kruse et al in Fig S12.

(Remarks on code availability)

NA

Reviewer #2

(Remarks to the Author)

The authors have addressed questions raised in the initial review and presented an improved manuscript. I recommend publication of this study.

(Remarks on code availability)

Reviewer #3

(Remarks to the Author)

The issues raised in the initial review have been satisfactorily addressed by the authors in their revised manuscript. There are still some minor issues such as incorrect references to supplementary figures that should be addressed by proofreading

before publication.

(Remarks on code availability)

All code and data is freely shared and provided in usable form. I only tested some random samples but didn't encounter any issue.

Reviewer #4

(Remarks to the Author)

(Remarks on code availability)

Comments made by reviewer #1

1) I think the data in Figure S1 and S2 is really important, and I suggest to move this to a main figure 1.

Our response – We moved the key parts of the Supplementary Fig. 1 to the main Fig. 1. The moved items include the synthesis scheme displaying modifications of the lead exemplar compound 1, structure of compound 21 (C-604) and the model of C-604 binding to the active site of GWL. Additionally, we included *in vitro* dose-response HTRF profiles for the L/R stereoisomers and the racemic mixture of C-604.

2) The model for how C-604 engages MASTL (Fig. S1C) have the authors analyzed this experimentally? If so, would be interesting to include.

Our response: Despite our best efforts, we could not obtain a crystal structure of the Greatwall Kinase domain with a resolution good enough to infer the mechanism of target engagement. However, the compound is an ATP-analogous structure that acts as an ATP competitor. Our model shows the most likely configuration of the small molecule inhibitor bound to the ATP binding pocket.

3) The authors observe inhibition of other kinases in the in vitro screens but claim that cellular phenotypes are not due to these kinases – it was a little unclear to me how they reached this conclusion. Could the please explain/expand.

Our response – We apologise for the lack of clarity in this section. To address this, we reworked the analysis in Figure S8, making it more robust and hopefully easier to understand. In addition to 0.5 μ M C-604 treatment, we assessed the impact of 2 μ M C-604 in cells transfected with siRNAs specific to suspected off-targets. We utilised several key features (derived from the imaging data) and UMAP dimensionality reduction to characterise each analysed cell population. The procedure is described in the new section “Greatwall inhibitor C-604 acts on-target” and the data are shown in Figure S9h, g and i.

4) In the phosphoproteomic analysis they compare their data with that of data from the Barr lab generated by a different approach and see little overlap. Have they compared to other PP2A-B55 data sets (Hein et al 2023, Kruse et al 2020, Swartz 2023)?

Our response – We expanded the analysis of PP2A-B55 substrates, including datasets from publications by Kruse et al. 2020 and Hein et al. 2023. The starfish meiosis study by Swartz et al. “Selective dephosphorylation by PP2A-B55 directs the meiosis I-meiosis II transition in oocytes,” was not included because of the lack of inter-species similarities of identified phospho-sites. The results of this analysis are shown in Figure S12, where we present a summary of high-confidence B55 substrates from comparisons between cell lines and previous phosphor-proteomic studies.

Comments made by reviewer #2

The main concern of the reviewer outlined in the initial paragraph are:

1) The novelty of a detailed cellular characterisation of Greatwall inhibition

“For example, a number of study by Dr. Jae-Sung Kim's group have produced enhanced GWL inhibitors that appear much more potent than the compound reported here, both in enzymatic assays and colony formation assays (PMID 34358073).”

Our response to point 1 – We acknowledge prior efforts to develop Greatwall kinase inhibitors, including work by our group, Dr. Kim's lab, and Pfizer. To date, the only other published study presenting cellular data on Greatwall inhibition is that of MKI-2 (PMID: 34358073). However, our data reveal distinct phenotypic differences between MKI-2 and our compound, C-604. MKI-2 induces a strong apoptotic response in breast cancer cells, whereas C-604 causes polyploidy and a cytostatic phenotype with a distinct gradient of sensitivity across various cell lines. To directly compare the two compounds, we synthesised MKI-2 and tested it in HeLa and RPE-1 cells. While MKI-2 inhibits ENSA phosphorylation at $\geq 0.5 \mu\text{M}$, it also exerts potent anti-proliferative effects at much lower concentrations ($<0.1 \mu\text{M}$), where no inhibition of ENSA(S67)/ARPP19(S62)-P is observed. This discrepancy suggests off-target effects at sub-micromolar doses. In contrast, C-604-induced phenotypes are B55 α -dependent and can be reverted by B55 α depletion, unlike MKI-2, whose effects are B55 α -independent. These findings imply that the cellular activity of MKI-2 likely stems from targets other than Greatwall kinase. Together, our comparative data underscore the novelty and importance of C-604 as a tool for dissecting the cellular functions of Greatwall kinase, establishing this study as the first to provide a detailed and selective cellular characterisation of its inhibition.

2) The novelty of suggesting that B55 levels are a predictive biomarker for sensitivity to Greatwall inhibition.

“The last point, that expression level of B55a determines MASTL inhibitor sensitivity, seems to be also covered by PMID 29976159: we found that MASTL-depletion-induced mitotic cell death was partially blocked by PP2A inhibition from PP2A-A α / β siRNA (Fig. 5i).”

We thank the reviewer for highlighting PMID: 29976159 and apologise for not citing it in the original manuscript. This study demonstrates that B55 α depletion partially reduces PARP cleavage upon MASTL knockdown in breast cancer cells. However, our study goes significantly beyond this observation. We propose and experimentally support a predictive model in which the ratio of B55 α to Greatwall kinase levels determines cellular sensitivity to Greatwall inhibition. Unlike the siRNA-based approach in PMID: 29976159, we use chemical inhibition in a panel of cell lines that display pronounced differences in sensitivity. We also experimentally modulate B55 α expression to test this model in U2OS cells. Furthermore, we now provide new data to validate the model across four additional CHOL cell lines, demonstrating consistent predictive value. This represents a substantial conceptual advance with potential translational implications for patient stratification in future clinical studies. We respectfully disagree that this mechanistic insight and its broader application are captured by Figure 5i of PMID: 29976159.

Further points by reviewer 2

1) The major argument is that different B55a levels in cancer cells determines sensitivity to GWL inhibitors (Figure 6E). However it appears that the overall trend received significant contribution from 2 cell lines (BT549, the most sensitive and HCC1143 the most resistant). If these two cell lines are taken out, It would be rather difficult to establish a trend using data from remaining cell lines. High reliance on two data points on the most extreme end is a concern.

Our response – The reviewer raises a valid concern. In the original manuscript, we attempted to rectify this by generating a panel of U2OS cell lines overexpressing different levels of B55 α . Predicted and experimental sensitivity to GWL inhibition correlated in 8 of 11 newly generated U2OS cell lines. To address the reviewer's comment and further test the predictive power of our model, we correlated predicted and experimental sensitivity to GWL inhibition in four additional cholangiocarcinoma cell lines, MMNK1, KKK-D068, KKK-D131 and KKU-055. Our model, based on B55 α and GWL expression levels, also predicted the sensitivity to GWL inhibition in these cell lines. We included the new data in Fig. 7c and d.

2) As a drug discovery paper, IC₅₀/EC₅₀ numbers should be clearly stated in main text.

Our response – We included *in vitro* IC₅₀ and cellular EC₅₀ values in the text. *In vitro* IC₅₀ values are also accompanied by the underlying HTRF-derived data for the racemic mixture and L- and R- stereoisomers of C-604. We included the new data in (Fig. 1), which now also contains the synthesis scheme, the chemical structure of C-604 and a model of C-604 binding to the GWL active site.

“According to the HTRF-based IC₅₀ evaluation, both stereoisomers (L, R) of compound 21 exhibited equivalent inhibitory potential (L/RIC₅₀ = 9.0 \pm 7.7 nM; LIC₅₀ = 10.9 \pm 2.0 nM; RIC₅₀ = 13.3 \pm 6.4 nM), demonstrating that compound 21 inhibited GWL in a non-stereospecific manner (Fig. 1c).”

“Importantly, estimated cellular EC₅₀ values (^{HCC1395}EC₅₀ = 0.28 \pm 0.21 μ M; ^{U2OS}EC₅₀ = 0.43 \pm 0.15 μ M; ^{RPE-1}EC₅₀ = 0.24 \pm 0.01 μ M; ^{HeLa}EC₅₀ = 0.31 \pm 0.20) displayed only minor, statistically insignificant variations, indicating that C-604 acted consistently in different cellular models (Supplementary Fig. 3a).”

3) Previous studies on similar conclusions e.g. PMID 29976159 should be cited.

Our response – We referred to the publication (PMID: 29976159) in the text and have tried to reference the literature on Greatwall kinase to the best of our knowledge and ability.

Comments made by reviewer #3

1) There seems to be limited correlation between the *in vitro* measured activity of the kinase panel (Fig. S1d) and the cellular IC₅₀ measurements of the selected candidates (Fig. S1e). The selection of kinases with >80% inhibition for further testing therefore seems arbitrary. There might well be additional kinases with cellular IC₅₀ < 50nM, which would need to be included in off target testing. The authors should expand the measurement of cellular IC₅₀ values to identify such candidates.

Our response: We identified most likely off-targets in two steps. (1) we tested the effect of 1 μ M C-604 on the *in vitro* activity of 100 kinases and selected the most impacted hits (> 80% inhibition). Since the high dose of C-604 (1 μ M) could be well beyond the point of inhibitory saturation, this experiment did not reveal anything about the efficiency of assessed C-604-mediated inhibition. (2) we further analysed the most likely C-604 off-targets and measured their dose-dependent C-604 responses and *in vitro* IC₅₀ values. Unfortunately, we did not measure cellular IC₅₀. Determined *in vitro* IC₅₀ values do not correlate with the maximal inhibition by C-604 (1 μ M) due to differences in the inhibitory potential of lower C-604 doses (< 100 nM). For instance, PRKD2 shows relatively high IC₅₀ (373 nM) because it is not affected by lower C-604 concentrations. In contrast, ULK2, which shows maximal inhibition lower than PRKD2, responds to C-604 even at lower concentrations (IC₅₀ = 21.6 nM). Since the *in vitro* IC₅₀ of C-604 against Greatwall is around 10 nM, we excluded off-target candidates with high *in vitro* IC₅₀ and focused the subsequent analysis on off-target suspects with IC₅₀ \leq 50 nM.

We agree that rigorous off-target assessment is essential. To this end, we have conducted a comprehensive analysis combining *in vitro* off-target screening, phenotypic validation, immunofluorescence studies, and mass spectrometry. In addition, we demonstrate functional rescue and synthetic synergy using siRNA targeting GWL and B55 α . Our phospho-proteomic analysis further indicates that the observed phenotypes are primarily due to B55-alpha reactivation following Greatwall inhibition. Of the 544 phosphorylation sites significantly reduced by C-604 in mitotic cells, 533 depend on B55 α . The remaining 11 include ENSA(S67), ARPP19(S62), the known GWL autophosphorylation sites (T873, S878), and nine additional sites without known function. Taken together, these data strongly support the selectivity of C-604 in cellular contexts.

2) This is specifically important as there are kinases with known mitotic phenotypes among the top 20 hits from the *in vitro* screening, such as the checkpoint kinase CHEK2 (PMID: 15048074, PMID: 35851582, PMID: 38559033). Kinases with known mitotic phenotypes should be tested for cellular IC₅₀ values and if applicable included in further testing.

Our response – We acknowledge the reviewer's concern regarding potential off-target effects on mitotic kinases identified in our *in vitro* screening, including CHEK2. However, our phosphoproteomic analysis in RPE-1 cells provides compelling evidence against significant off-target contributions. Of the 544 phosphorylation sites reduced by C-604, 533 are strictly dependent on B55 α , indicating that their dephosphorylation is downstream of PP2A-B55 activation rather than direct kinase inhibition. Direct Greatwall substrates (ENSA S67, ARPP19 S62, GWL T873/S878) remain significantly affected even in B55 α -depleted cells, serving as

positive controls for direct kinase inhibition. In contrast, the 11 phosphorylation sites that remain altered in B55 α -depleted cells are not known targets of CHEK2 or other well-characterized mitotic kinases, and none are associated with established CHEK2-regulated pathways. While we agree that further cellular IC50 testing may be informative for selected kinases, the available data strongly support that C-604's cellular effects are primarily mediated via selective Greatwall inhibition and PP2A-B55 reactivation.

3) Testing for off target effects using phenotypic reversal upon siRNA is overall convincing. However, the authors performed these tests only at 0.5 μ M C-604 concentration (Fig. S7c-d), while most of the phenotypic studies for GWL inhibition were done at up to 2 μ M (Fig. 1-4). As the observed effects were most obvious at high concentrations, it would be advisable to repeat the off-target kinase testing at 2 μ M C-604 as well. Also, some knockdowns show strong cell cycle alterations already in control condition (e.g. NUAK, MELK and CHEK1 siRNA abolishing S-phase populations in RPE-1 cells). The author should comment to which extent these alterations affect conclusions about phenotypic outcomes of GWL inhibition. If cells are not dividing, mitotic defects will not be assessable.

Our response – We agree with reviewer #3 that this analysis was relatively limiting and unintuitive, as also suggested by reviewer #1. As per suggestion, we included the effects of 2 μ M C-604 and reworked the analysis. Our new analysis characterises differences and similarities between individual cell populations using several key features (derived from the imaging data), UMAP-based dimensionality reduction and Wasserstein distances between distinct UMAP projections. The new section, “Greatwall inhibitor C-604 acts on-target” and the new Figure S8, includes the logic behind this procedure and detailed description of individual steps. We believe that the strong effects of siRNAs specific to NUAK, MELK and CHEK1 are very different from those induced by C-604, implying different molecular causes. Consequently, NUAK, MELK and CHEK1 are unlikely to be genuine off-targets.

4) The authors suggest that the phenotypic outcome of GWL inhibition is only partially dependent on p53 (page 8, line 205-214). This finding is interesting and relevant. However, its validity is hard to assess as the data in Figure S5 is missing. Instead, the figure panels shown are a replication of Fig. 5. Moreover, the authors should further discuss which p53-independent pathways could be involved in the physiological response to GWL inhibition.

Our response – We made a mistake and incorrectly replicated Fig. 5 instead of supplying Supplementary Fig. 5. We thank the reviewer for pointing out this error and have corrected this mistake. Supplementary Fig. 5 shows genetic alterations impacting p53 function in relevant cell lines and single cell representation of immunofluorescence microscopy data (RPE-1 WT and p53 cells) treated with 0 or 2 μ M C-604). To further address this observation, we discuss the hypothetical involvement of the p53 family protein p73, which has been implicated in the suppression of polyploidy and aneuploidy independently of p21 stabilisation (PMIDs: 18805989, 17707235). Publishing this observation is meant to function as a precursor for a follow-up publication addressing this issue in adequate detail.

5) The statistical analysis of the data should be reviewed as the authors apply t-test for all data sets. The applicability of t-test should be confirmed by assessing normality of the underlying distributions, and corrections for multiple testing should be applied where necessary. It might be helpful to include analysis of effect sizes, e.g. by calculating Cohen's d values.

Our response – Following the suggestions by the reviewer, we have reviewed our statistical analysis. We used the Shapiro–Wilk test to assess the normality of the data. In cases where parametric testing could not be applied, we used the Wilcoxon signed-rank test. We corrected determined p-values using the Benjamini-Hochberg procedure in analyses involving multiple statistical tests. For data presented in (Fig. 2), we report the results of the Shapiro–Wilk test and effect sizes (Cohen's d values) as independent graphs (updated Supplementary Fig. 4c, d). Otherwise, Shapiro–Wilk test results (the minimum recorded p-values) are included in figure legends.

6) To allow evaluation of the dependency for PP2A-B55 α , the extent and specificity of the knockdown of all subunits should be shown, optimally at the protein level (Fig. 3).

To assess isoform-specific knockdown efficiency, we performed RT-qPCR for B55 α , B55 β , B55 γ , and B55 δ following siRNA transfection. Significant reductions were observed in B55 α , B55 β , and B55 δ transcripts, while B55 γ expression was undetectable across all conditions, consistent with its known brain-specific expression profile. These results are presented in Supplementary Fig. 3d. We attempted to evaluate protein-level knockdown using commercial antibodies; however, the B55 β antibody (Abcam ab251885) showed cross-reactivity with other isoforms, and the B55 γ antibody (Santa Cruz sc-100417) exhibited strong non-specific binding. Likewise, antibodies for B55 δ (e.g. and N2C3) showed strong cross-reactivity with B55 α . This is not surprising, given the high similarity of the different paralogues at the amino-acid level. Despite these technical limitations, the mRNA-level data provide robust evidence for isoform-specific knockdown.

7) The model for predicting sensitivity to GWL inhibition should be experimentally tested with cell lines not used for model generation (Fig. 6).

Our response – In the original manuscript, we tested the model by assessing the C-604 sensitivity in 11 clonal U2OS cell lines with different expression levels of B55 α . We predicted the sensitivity of 8 out of 11 of these cell lines, indicating a strong predictive power. We understand the concern that the training cell line panel included the parent U2OS (WT) cell line. To rectify this and further test the credibility of our model, we assessed the cellular responses of independent cholangiocarcinoma cell lines KKK-D068, KKK-D131, KKK-055 and MMNK1. This analysis further supported our predictive model. We included the new dataset alongside the original contents in Fig. 7.

8) Please ensure that all experimental procedures are fully described in the methods section. For example, currently only sequences of siRNAs are given, but no information about transfection procedures and concentrations used. Also, please expand legends to include all important information (assays and conditions used) and explain normalization and visual display of the data. Some legends miss descriptions of entire figure panels (e.g. Fig. 8d-f)!

Our response – We included the siRNA transfection protocol in the methods section. We enriched the figure legends to provide more detail about the presented data. We included the (Fig. 8d-f) description.

Minor points:

The initial screen of the AstraZeneca compound collection is described in the first paragraph of the results section, but it is hard to assess without any representation of the data. For example, six clusters are mentioned from which compound 1 emerged as a candidate without giving any information about them.

Our response – We included the characterisation of the initial compound clusters. This dataset is discussed in the section: “Development of the novel GWL kinase inhibitor C-604”. Graphical representation of this dataset is in (Supplementary Fig. 1).

Could the authors explain why two Conditional Degron Tags (mAID and SMASh) were used to degrade PP2A-B55 α ?

Our response – To clarify the utility of double degron tagging we introduced the following sentence: “We tagged B55 α with two complementary degron motifs, mAID (PMID: 19915560) and SMASh (PMID: 26214256), as the combination of these degrons improves degradation efficiency and minimises the residual levels of targeted protein (PMID: 30008317).”

Please add information of how many cells were analyzed for all data sets (missing e.g. for Figure 2d, Figure 3, Figure S6).

Our response – For the sake of readability, we included minimum and maximum number of analysed cells in respective figure legends. To provide a complete set of analysed cell counts, we generated a set of .csv files, which can be obtained from the figshare repository published alongside this work.

Please make sure that abbreviations are spelled out at first time of usage, which might be as part of a supplemental data figure, to help readers not familiar with these kind of measurements / analysis.

Our response – All abbreviations are spelled out the first time they are used in the updated manuscript.

Figure 3b: It might be helpful to not only statistically test changes of cell cycle distributions to the 0 μ M condition for each siRNA, but also compare the corresponding conditions to the siCTR condition.

Our response – We included these statistical comparisons alongside Cohen's d values (Supplementary Fig. 6c).

Figure 4d: Euclidean distance was calculated in Figure 2g, so the used Euclidean values correspond to Figure 2g, not Figure 3f.

Our response – We corrected the mistake in Fig. 4d description: "Correlation scatterplots of defective mitotic exit frequencies and magnitudes of cellular responses to C-604 treatment (Fig. 2g)".

Figure 4g/h: As values seem discrete, could it be that median, not means are represented by red lines?

Our response – The reviewer is right, we made a mistake and referred to means instead of medians. We corrected the figure description: "(g, h) Red lines represent medians".

Figure 4f: Please add units for the concentration of C-604.

Our response – We added the C-604 concentration units to Fig. 4f.

Figure 5 e and f: Legend should explain difference between Figure 5e and Figure 5f.

Our response – We changed the Fig. 5e,f description to emphasise that two figures display enrichments of biological process and cellular component gene ontology terms: " Biological process (e) and cellular component (f) gene ontology terms significantly enriched in phosphopeptides dephosphorylated in prometaphase RPE-1 B55 α -dd (-DIA) and HeLa cells treated with C-604."

Figure S1b: It would be nice to show the experimental data used to estimate IC₅₀ values as in Fig. S1e.

Our response – We include the HTRF data for C-604 stereoisomers and the racemic mixture in the updated Fig. 1c. The IC₅₀ values are slightly different to those reported in the original manuscript. This is because the original values were calculated using several different ranges of C-604 concentrations. We decided to only use the data from experiments using the same range. These differences do not change any conclusions.

Figure S2a: The choice of concentrations seems suboptimal for generating a valid fit. Further information about how the in vitro assay was performed and how the phosphorylation of ENSA was finally measured would be helpful for the reader.

Our response – We agree that this dataset is of sub-optimal quality. Consequently, we decided to remove it from the manuscript. We present the HTRF-based assessment of C-604 IC₅₀

instead. We believe this alteration does not impact the quality of *in vitro* C-604 characterisation.

Lines 117-124: The reference to Figures S1 d,e,f is missing.

Our response – We added the missing references.

Line 216: The reference to Figure S2d is wrong, it should be Figure S2e.

Our response – We corrected the reference.

Lines 228-229: The reference to Figure S2d is wrong, it should be Figure S2e.

Our response – We corrected the reference.

Line 240: The reference to Figure 5S is wrong, it should be Figure S6.

Our response – We corrected the reference.

Please add page numbers in a revised manuscript.

Our response – We added the page numbers.

The code was only briefly reviewed and seemed reasonable. On first glance, I didn't find the respective data files, so I could not reproduce the analysis.

Our response – We created a figshare repository containing all relevant datasets. Shared datasets can be downloaded using the private “figshare” links. Registered DOIs are currently inactive. The Github repository containing the R scripts used in the analysis has been updated. The .readme file now also contains links to relevant datasets.

Dataset	Repository
Phosphoproteomics	https://doi.org/10.6084/m9.figshare.28829927 https://figshare.com/s/9883aaaa9e28d1bbbffd
Immunofluorescence microscopy	https://doi.org/10.6084/m9.figshare.28829816 https://figshare.com/s/c337d11f90e297097735
Live-cell imaging	https://doi.org/10.6084/m9.figshare.28829915 https://figshare.com/s/089f3af9dec771190ccc
Colony formation	https://doi.org/10.6084/m9.figshare.28829903 https://figshare.com/s/e3ab94eba5f1c893f0e4
Western blot	https://doi.org/10.6084/m9.figshare.28829306 https://figshare.com/s/155fa64c8a8b8d2474c4
TCGA	https://doi.org/10.6084/m9.figshare.28848074 https://figshare.com/s/36a08998b59b142a2548

(Reviewer #1) Small point is that Krus et al should be Kruse et al in Fig S12.

Response – We corrected this.

(Reviewer #3) The issues raised in the initial review have been satisfactorily addressed by the authors in their revised manuscript. There are still some minor issues, such as incorrect references to supplementary figures, that should be addressed by proofreading before publication.

Response – We corrected the incorrect references to Supplementary Figures.